# RNA-Protein Interaction Analysis of SARS-CoV-2 5′ and 3′ Untranslated Regions Reveals a Role of Lysosome-Associated Membrane Protein-2a during Viral Infection

Rohit Verma,[a] Sandhini Saha,[b] Shiv Kumar,[a] Shailendra Mani,[c] Tushar Kanti Maiti,[b] Milan Surjit[a]

[a]Virology Laboratory, Translational Health Science and Technology Institute, NCR Biotech Science Cluster, Faridabad, Haryana, India
[b]Laboratory of Functional Proteomics, Regional Centre for Biotechnology, NCR Biotech Science Cluster, Faridabad, Haryana, India
[c]Translational Health Science and Technology Institute, NCR Biotech Science Cluster, Faridabad, Haryana, India

Rohit Verma and Sandhini Saha contributed equally to this article. Author order was determined both alphabetically and in order of increasing seniority.

**ABSTRACT** Severe acute respiratory syndrome coronavirus 2 (SARS-CoV-2) is a positive-strand RNA virus. The viral genome is capped at the 5′ end, followed by an untranslated region (UTR). There is a poly(A) tail at the 3′ end, preceded by a UTR. The self-interaction between the RNA regulatory elements present within the 5′ and 3′ UTRs and their interaction with host/virus-encoded proteins mediate the function of the 5′ and 3′ UTRs. Using an RNA-protein interaction detection (RaPID) assay coupled to liquid chromatography with tandem mass spectrometry, we identified host interaction partners of SARS-CoV-2 5′ and 3′ UTRs and generated an RNA-protein interaction network. By combining these data with the previously known protein-protein interaction data proposed to be involved in virus replication, we generated the RNA-protein-protein interaction (RPPI) network, likely to be essential for controlling SARS-CoV-2 replication. Notably, bioinformatics analysis of the RPPI network revealed the enrichment of factors involved in translation initiation and RNA metabolism. Lysosome-associated membrane protein-2a (Lamp2a), the receptor for chaperone-mediated autophagy, is one of the host proteins that interact with the 5′ UTR. Further studies showed that the Lamp2 level is upregulated in SARS-CoV-2-infected cells and that the absence of the Lamp2a isoform enhanced the viral RNA level whereas its overexpression significantly reduced the viral RNA level. Lamp2a and viral RNA colocalize in the infected cells, and there is an increased autophagic flux in infected cells, although there is no change in the formation of autophagolysosomes. In summary, our study provides a useful resource of SARS-CoV-2 5′ and 3′ UTR binding proteins and reveals the role of Lamp2a protein during SARS-CoV-2 infection.

**IMPORTANCE** Replication of a positive-strand RNA virus involves an RNA-protein complex consisting of viral genomic RNA, host RNA(s), virus-encoded proteins, and host proteins. Dissecting out individual components of the replication complex will help decode the mechanism of viral replication. 5′ and 3′ UTRs in positive-strand RNA viruses play essential regulatory roles in virus replication. Here, we identified the host proteins that associate with the UTRs of SARS-CoV-2, combined those data with the previously known protein-protein interaction data (expected to be involved in virus replication), and generated the RNA-protein-protein interaction (RPPI) network. Analysis of the RPPI network revealed the enrichment of factors involved in translation initiation and RNA metabolism, which are important for virus replication. Analysis of one of the interaction partners of the 5′-UTR (Lamp2a) demonstrated its role in reducing the viral RNA level in SARS-CoV-2-infected cells. Collectively, our study provides a resource of SARS-CoV-2 UTR-binding proteins and identifies an important role for host Lamp2a protein during viral infection.

Address correspondence to Milan Surjit, milan@thsti.res.in.

**KEYWORDS** SARS-CoV-2, 5′ UTR, 3′ UTR, RNA-protein interaction network, coronavirus, virus-host interaction, Lamp2, Lamp2a

In December 2019, a highly pathogenic coronavirus was identified in the city of Wuhan, China, and was named 2019-nCoV/SARS-CoV-2 (1–3). Since then, the virus has spread globally and the World Health Organization (WHO) has classified the outbreak as a pandemic. Severe acute respiratory syndrome coronavirus 2 (SARS-CoV-2) belongs to the family *Coronaviridae*. Coronaviruses are known to be present in animals and humans for a long time, usually resulting in respiratory and intestinal dysfunction in the host (4). Until the emergence of SARS-CoV in 2002, coronaviruses were not considered to be a significant threat to human health (5). However, within the last 18 years since the SARS outbreak, three major human coronavirus outbreaks have occurred, suggesting that highly pathogenic human coronaviruses are evolving quickly. Considering the severity of the current SARS-CoV-2 pandemic, there is an urgent need to understand the life cycle and pathogenetic mechanism of the virus. Since all coronavirus genomes share significant homology, the knowledge obtained from the study of the SARS-CoV-2 genome will be useful to formulate a long-term action plan to deal with the current as well as future coronavirus outbreaks.

Coronaviruses are positive-strand RNA viruses. The viral genome serves as the template for synthesis of antisense strand and production of proteins involved in replication as well as assembly of the progeny virions (6). The viral genome is capped at the 5′ end and polyadenylated at the 3′ end. 5′ and 3′ ends of the genome contain noncoding sequences, also known as untranslated regions (UTRs). 5′- and 3′ UTRs in the positive strand of RNA viruses play essential regulatory roles in virus replication, enhancing the stability of the viral genomic RNA, host immune modulation, and encapsidation of the viral genome into the nucleocapsid core. Additionally, *cis*-acting regulatory elements are present within the coding regions of the positive and negative strand (replication intermediate) of the viral genome. These regulatory RNA elements also play significant roles in the life cycle and pathogenesis of the virus. Intraviral interactions between regulatory RNA elements of the virus and intraviral as well as virus-host RNA-protein interactions control the function of the 5′ and 3′ UTRs and internal *cis*-acting RNA elements of the virus (7, 8).

SARS-CoV-2 contains a 265-nucleotide-long 5′ UTR and a 228-nucleotide-long 3′ UTR. These UTRs show considerable homology with the 5′ and 3′ UTRs of other betacoronaviruses such as SARS and SARS-related betacoronaviruses (9). Distinct stem-loops and secondary structures within the UTRs are known to mediate their regulatory function. Notably, stem-loop I and stem-loop II (SL-I and SL-II) of the 5′ UTR are important for long-range interaction and subgenomic RNA synthesis, respectively. SL-III contains the translation regulatory sequence (TRS), which is essential for the discontinuous transcription of ORF1ab. SL-5 is crucial for the viral RNA packaging and translation of the ORF1ab polyprotein (10–12).

The 3′ UTR of coronaviruses is important for viral replication (RNA synthesis and translation). It contains a hypervariable region (HVR) which has been shown to be important for pathogenesis of mouse hepatitis virus (MHV). HVR contains a stem-loop II-like motif (S2M) that associates with host translation factors (13). The S2M motif is also present in the 3′ UTR of SARS-CoV-2 (9).

Interaction of host proteins with the 5′ and 3′ UTRs of viral genomic RNA is important for the replication and pathogenesis of many RNA viruses. Interaction of polypyrimidine tract binding protein (PTB) with the 5′ UTR-TRS of MHV is known to control transcription of the viral RNA (14–16). Furthermore, RNA helicases such as DDX1 and DHX15, proteins involved in translation regulation such as eIF1$\alpha$ and eIF3S10, and proteins involved in cytoskeleton movement such as tubulin, Annexin A2, moesin, and GAPDH (glyceraldehyde 3-phosphate dehydrogenase) are known to associate with the 5′ UTR of a few coronaviruses (7, 17). The 3′ UTR of MHV associates with hnRNPA1 and modulates viral replication (18). Poly(A) binding protein (PABP), transcriptional

mSystems®

activator p100 (SND1), and heat shock proteins (HSP40, HSP60, HSP70) associate with the 3′ UTR of MHV (19). The functional significance of some of these interactions is known (8).

Despite general acceptance of the importance of the 5′ and 3′ UTRs of RNA viruses in controlling their replication and pathogenesis, no systematic study has been undertaken to elucidate the molecular composition of the RNA-protein complex assembled at the 5′ and 3′ UTRs of SARS-CoV-2 and the function of 5′ and 3′ UTRs of SARS-CoV-2.

We employed an RNA-protein interaction detection (RaPID) assay coupled to liquid chromatography with tandem mass spectrometry (LC-MS/MS) to identify the repertoire of host proteins that interact with the SARS-CoV-2 5′ and 3′ UTRs (20). These data sets were used to construct the virus-host RNA-protein-protein interaction (RPPI) network. *In silico* analyses of the RPPI network revealed enrichment of proteins involved in multiple processes, such as Cap-dependent translation and RNA metabolism. Further studies revealed an antiviral role of LAMP2a during SARS-CoV-2 infection. The functional significance of these findings in SARS-CoV-2 replication and pathogenesis is discussed.

## RESULTS

**Identification of host proteins that interact with the 5′ and 3′ UTRs of SARS-CoV-2 genomic RNA.** Three hundred nucleotides from the 5′ end (designated the 5′-UTR RNA) and 203 nucleotides from the 3′ end (3′-UTR RNA) of the SARS-CoV-2 genomic RNA (Wuhan isolate), which includes the 5′ and 3′ UTRs (nucleotides 1 to 265 and 29676 to 29878, respectively, in the viral genome), were cloned into the pRMB vector between the BirA ligase binding stem-loop (SL-A and SL-B) sequences (Fig. 1A to C). 5′ and 3′ UTRs consist of 265 and 228 nucleotides, respectively. Extra bases were included at the 5′ end to ensure that stem-loops in the predicted secondary structures of the 5′ UTR remain intact in the hybrid RNA (Fig. 1A). Eight adenine residues were retained at the 3′ end (Fig. 1B). An mfold-mediated comparison of the secondary structures of the 5′ 300-nucleotide and 3′ 203-nucleotide RNA sequences with and without BirA ligase binding stem-loop RNA sequences indicated that secondary structures of viral 5′ UTR and 3′ UTR and BirA binding stem-loops (SL-A and SL-B) are not disturbed in the hybrid RNA sequence (Fig. 1A and B; see Fig. S1 and S2 in the supplemental material). A RaPID (RNA-protein interaction detection) assay was performed to identify the host proteins that interact with the 5′-UTR and 3′-UTR RNA in HEK293T cells. The RaPID assay allows BirA ligase-mediated *in vivo* biotin labeling of host proteins that interact with the RNA sequence of interest cloned between the BirA ligase-binding RNA motifs. Biotinylated proteins are captured using streptavidin agarose beads and identified by LC-MS/MS (Fig. 1C) (20). Thus, both transiently interacting and stably interacting proteins can be identified using this technique.

Expression of BirA ligase in HEK293T cells was confirmed by Western blotting using anti-hemagglutinin (anti-HA) antibody that recognizes the HA tag fused to the BirA ligase (Fig. 1D). The optimal duration of biotin treatment was selected through a time course analysis of biotin labeling, based on which an 18-h labeling period was selected (Fig. 1E). To test the functionality of the RaPID assay, a known RNA-protein interaction was tested. We measured the interaction between a UG-rich RNA sequence called EDEN15 and CUGBP1 protein (CUG triplet repeat, RNA binding protein 1) by the RaPID assay as a positive control because an earlier study demonstrated their interaction by RaPID assay (20). Streptavidin pulldown of EDEN15 RNA-interacting proteins, followed by Western blotting using CUGBP1 antibody, revealed the interaction between them, in agreement with an earlier report and confirming the validity of the technique under our laboratory condition (Fig. 1F) (20).

HEK293T cells expressing the SL-A–5′ UTR–SL-B or SL-A–3′ UTR–SL-B or SL-A–SL-B (pRMB) were treated with biotin, followed by pulldown of biotinylated proteins and LC-MS/MS analysis. Biotin-untreated cells were processed in parallel as a control. Samples from three independent experiments were run in triplicate. Pearson correlation analysis of MS data demonstrated good correlation between replicates for individual biological samples (average range, 0.5 to 0.98) (Fig. 2A). Specific and strong

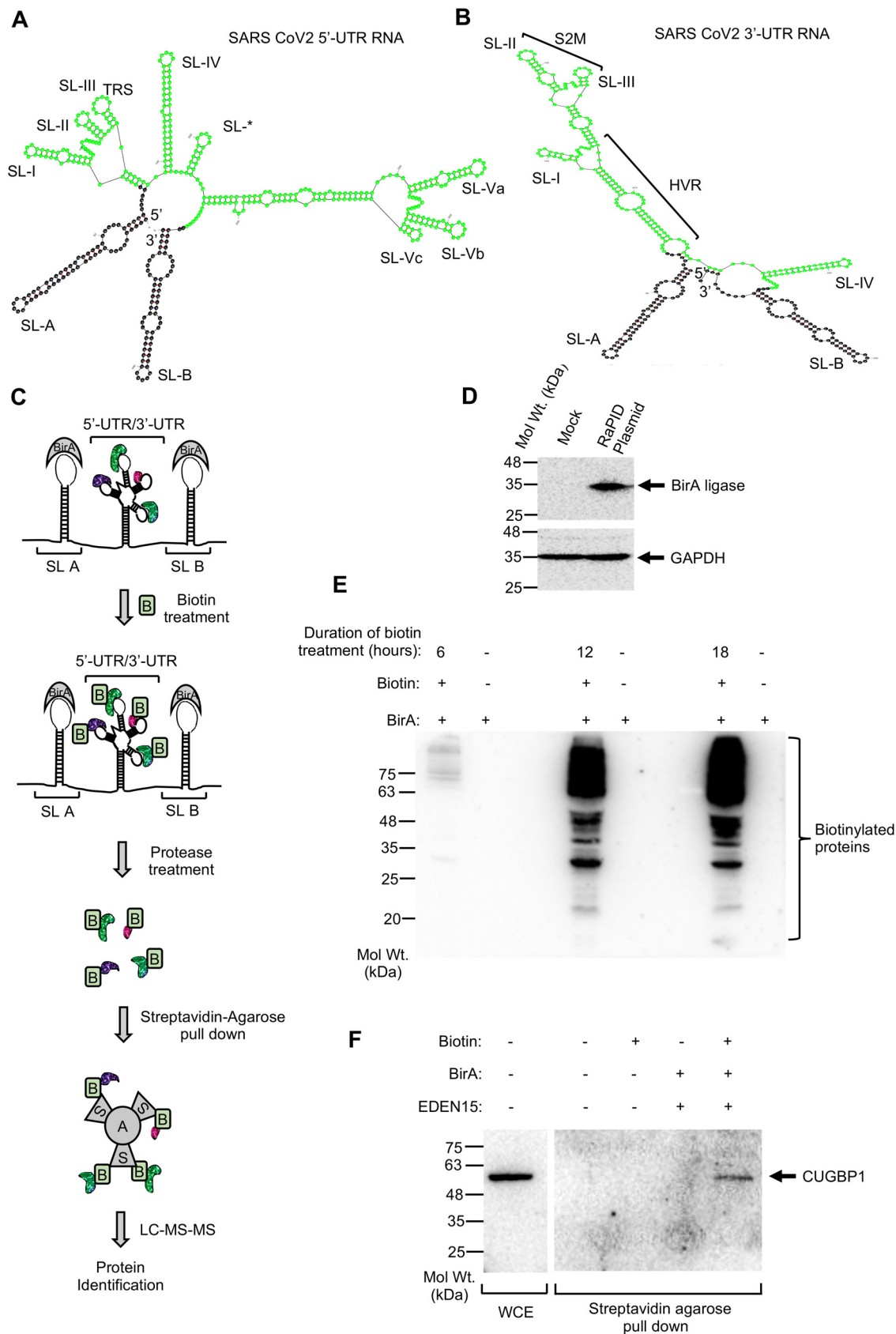

**FIG 1** Establishment of RaPID assay to identify the host interaction partners of SARS-CoV-2 5′-UTR and 3′-UTR RNA. (A) Schematic of predicted secondary structure of SARS-CoV-2 5′-UTR RNA. An asterisk denotes an unannotated stem-loop in the 5′ UTR. SL-A

interaction partners of 5'-UTR and 3'-UTR RNAs were identified in three steps: (i) only those proteins having at least one biotinylated peptide and a posterior error probability (PEP) score of 15 or more in all LC-MS samples were selected (Fig. S3; Table S1) (21); (ii) proteins were selected after subtraction of pRMB data set from 5'-UTR and 3'-UTR data sets (Fig. 2B); (iii) from the background-subtracted data set, only those proteins with a minimum of two unique peptides and a "prot score" of 40 or more were considered for further analysis (Table 1) (see Materials and Methods).

By using a similar protocol, in an unrelated study, we had identified host interaction partners of the 3' end (3' UTR plus adjacent 100 nucleotides) of hepatitis E virus (HEV) genomic RNA (Table S2). HEV is a positive-strand RNA virus of the *Hepeviridae* family, and the 3' end of HEV harbors a secondary structure composed of multiple stem-loops that have been predicted to be important for viral replication (22). In order to further increase our confidence in the reliability of the RaPID technique, the HEV 3'-end RNA-interacting protein list was compared to the SARS-CoV-2 5'-UTR and 3'-UTR RNA-interacting protein list. One protein (histone H3.1) was found to be in common for the two data sets, indicating specificity of the interacting proteins for the RNA baits. Therefore, using the RaPID assay, 47 and 14 host proteins were identified to interact with the 5'-UTR RNA and 3'-UTR RNA, respectively (Table 1). Out of these, 4 proteins (UBA1, GNL2, JAKMIP3, and YTHDF3) interacted with both 5' and 3' UTRs (Table 1, represented in bold font). A search in the Human Protein Atlas data bank revealed that 55 of 57 host proteins identified in the RaPID assay are expressed in both lungs and intestine of human (Table S3).

Recently, two independent studies identified the RNA-protein interactome of the SARS-CoV-2 genome by use of the RAP-MS (RNA antisense purification with mass spectrometry) and ChIRP-MS (comprehensive identification of RNA-binding proteins by mass spectrometry) techniques (23, 24). A comparison with those data identified 27 proteins to be common to the three data sets (Table 2). Of those proteins, 11 are common between RaPID and RAP-MS data and 25 proteins are common between RaPID and ChIRP-MS data (Table 2). The majority of common proteins are bound to the 5' UTR (23 proteins).

Recent studies have also identified a number of host factors that are important for SARS-CoV-2 infection (24–26). In order to gain a functional insight into the RaPID-identified host protein data set, we next searched for host proteins that are common between the RaPID data and the CRISPR knockout screening data. Six SARS-CoV-2 5'-UTR RNA-binding host proteins were found to be present in the CRISPR knockout screening data set reported by Daniloski et al. (25). One of the 5'-UTR binding proteins (TPR) identified to be important for SARS-CoV-2 infection in the study of Daniloski et al. was also found to be present in the CRISPR knockout screening data set reported by Wei et al. (26). Nine proteins were found to be present in the CRISPR knockout screening data set reported by Flynn et al. (25), of which 7 proteins were bound to the 5'-UTR RNA and 2 proteins were bound to the 3'-UTR RNA (Table 2). Of the 9 proteins, only G3BP2 displayed proviral characteristics, whereas the remaining 8 proteins showed antiviral characteristics. Collectively, these analyses support an important role of the RaPID-identified host proteins in the SARS-CoV-2 infection process.

**Construction and analysis of the RPPI network at the 5' and 3' ends of the SARS-CoV-2 genome.** 5'-UTR and 3'-UTR RNA binding protein data sets were imported to Cytoscape to construct the RNA-protein-protein interaction (RPPI) network

**FIG 1** Legend (Continued)

and SL-B represent BirA-binding RNA motifs, TRS denotes the transcriptional regulatory sequence, and SL-I to SL-V denote the stem-loops present in the 5'-UTR RNA. (B) Schematic of predicted secondary structure of SARS-CoV-2 3'-UTR RNA. SL-A and SL-B represent BirA-binding RNA motifs, HVR denotes the hypervariable region, S2M denotes the stem-loop II-like motif, and SL-I to SL-IV denote the stem-loops present in the 3'-UTR RNA. (C) Schematic of RaPID assay workflow. B, biotin; A, agarose; S, streptavidin. (D) Western blot detection of BirA ligase and GAPDH level in HEK293T cells transfected with the RaPID plasmid for 48 h. (E) Western blot detection of biotinylated protein in HEK293T cells transfected with BirA ligase and treated with biotin for different times, as indicated. (F) Western blot detection of CUGBP1 protein in the whole-cell extract of HEK293T cells (WCE) and in streptavidin agarose pulldown of EDEN15 RNA-interacting biotinylated proteins.

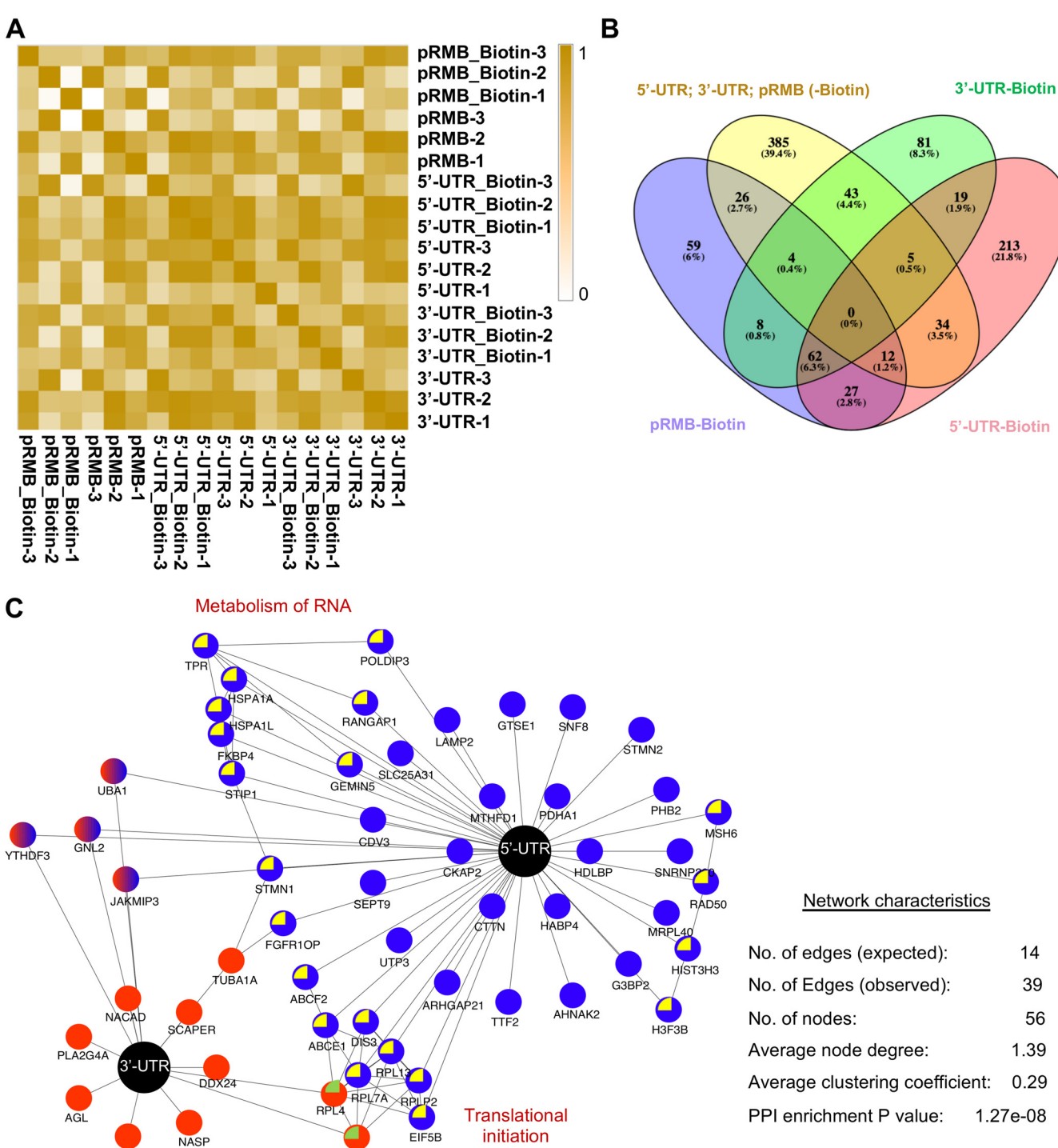

**FIG 2** Identification of SARS-CoV-2 5′-UTR and 3′-UTR RNA-interacting host proteins by RaPID assay. (A) Clustering analysis of correlation between biological replicates used in LC-MS/MS. (B) Venny analysis of 5′-UTR and 3′-UTR RNA-interacting proteins. (C) Schematic of RNA-protein interaction network of the 5′-UTR + 3′-UTR SARS-CoV-2 RNAs. Black nodes, 5′ UTR or 3′ UTR; blue or red nodes, host protein. Host proteins that interact with each other are indicated by yellow or green color inside the node. Common host proteins that interact with both 5′-UTR and 3′-UTR RNA are represented by dual (red and blue) colors.

(27). Analysis of the network parameters such as edge (number of interactions), node (protein), node degree (number of interactions linked to a protein), and clustering coefficient (tendency of nodes to remain in a cluster) revealed that 3′-UTR interacting proteins did not show any significant network characteristics (Fig. S3B). Note that a PPI network is considered significant when the observed network connectivity is

**TABLE 1** Host proteins that interact with 5′-UTR and 3′-UTR RNA of SARS-CoV-2, identified by RaPID assay

| Host protein type and gene name[a] | Description | Prot score | No. of biotinylated peptides | No. of unique peptides |
|---|---|---|---|---|
| SARS-CoV-2 5′-UTR RNA-interacting host proteins | | | | |
| HSPA1L | Heat shock 70-kDa protein 1-like | 492 | 17 | 10 |
| **UBA1** | Ubiquitin-like modifier-activating enzyme 1 | 380 | 6 | 4 |
| MTHFD1 | C-1-tetrahydrofolate synthase, cytoplasmic | 372 | 21 | 8 |
| HSPA1A | Heat shock 70-kDa protein 1A/1B | 278 | 10 | 7 |
| HIST3H3 | Histone H3.1t | 212 | 10 | 3 |
| HIST1H3A | Histone H3.1 | 212 | 9 | 3 |
| H3F3A | Histone H3.3 | 212 | 12 | 3 |
| DIS3 | Exome complex exonuclease RRP44 | 180 | 18 | 7 |
| PHB2 | Prohibitin-2 | 132 | 10 | 4 |
| RPL7A | 60S ribosomal protein L7a | 120 | 29 | 13 |
| LAMP2 | Lysosome-associated membrane glycoprotein 2 | 119 | 4 | 2 |
| RPLP2 | 60S acidic ribosomal protein P2 | 110 | 4 | 2 |
| SNF8 | Vacuolar-sorting protein SNF8 | 104 | 7 | 3 |
| G3BP2 | Ras GTPase-activating protein-binding protein 2 | 98 | 4 | 2 |
| ABCE1 | ATP-binding cassette subfamily E member 1 | 79 | 10 | 6 |
| RPL13 | 60S ribosomal protein L13 | 78 | 7 | 4 |
| CTTN | Src substrate cortactin | 70 | 7 | 6 |
| ABCF2 | ATP-binding cassette subfamily F member 2 | 70 | 3 | 2 |
| TPR | Nucleoprotein TPR | 65 | 22 | 18 |
| CKAP2 | Cytoskeleton-associated protein 2 | 63 | 19 | 13 |
| SNRNP200 | U5 small nuclear ribonucleoprotein | 61 | 23 | 16 |
| EIF5B | Eukaryotic translation initiation factor 5B | 61 | 30 | 12 |
| SLC25A31 | ADP/ATP translocase 4 | 60 | 12 | 7 |
| PDHA1 | Pyruvate dehydrogenase E1 component subunit alpha, somatic form, mitochondrial | 60 | 12 | 5 |
| **GNL2** | Nucleolar GTP-binding protein 2 | 55 | 13 | 10 |
| GEMIN5 | Gem-associated protein 5 | 55 | 5 | 3 |
| **JAKMIP3** | Janus kinase and microtubule-interacting protein 3 | 54 | 7 | 5 |
| GTSE1 | G$_2$ and S phase-expressed protein 1 | 54 | 9 | 4 |
| MSH6 | DNA mismatch repair protein Msh6 | 52 | 15 | 9 |
| STIP1 | Stress-induced phosphoprotein 1 | 52 | 13 | 5 |
| SEPT9 | Septin-9 | 50 | 10 | 4 |
| STMN2 | Stathmin-2 | 49 | 8 | 6 |
| FGFR1OP | FGFR1 oncogene partner | 49 | 11 | 4 |
| STMN1 | Stathmin | 49 | 6 | 4 |
| MRPL40 | 39S ribosomal protein L40, mitochondrial | 49 | 4 | 3 |
| RANGAP1 | Ran GTPase-activating protein 1 | 49 | 8 | 3 |
| TTF2 | Transcription termination factor 2 | 48 | 7 | 3 |
| **YTHDF3** | YTH domain family protein 3 | 47 | 3 | 3 |
| HDLBP | Vigilin | 46 | 30 | 14 |
| RAD50 | DNA repair protein RAD50 | 46 | 21 | 13 |
| AHNAK2 | Protein AHNAK2 | 45 | 101 | 47 |
| POLDIP3 | Polymerase delta-interacting protein 3 | 45 | 8 | 5 |
| ARHGAP21 | Rho GTPase-activating protein 21 | 41 | 19 | 15 |
| UTP3 | Something about silencing protein 10 | 41 | 5 | 4 |
| CDV3 | Protein CDV3 homolog | 41 | 3 | 3 |
| HABP4 | Intracellular hyaluronan-binding protein 4 | 41 | 7 | 2 |
| FKBP4 | Peptidyl-prolyl *cis-trans* isomerase FKBP4 | 40 | 7 | 4 |
| SARS-CoV-2 3′-UTR RNA-interacting host proteins | | | | |
| TUBA1A | Tubulin alpha-1A chain | 536 | 5 | 4 |
| **UBA1** | Ubiquitin-like modifier-activating enzyme 1 | 395 | 4 | 3 |
| NASP | Nuclear autoantigenic sperm protein | 218 | 3 | 3 |
| RPL4 | 60S ribosomal protein L4 | 80 | 2 | 2 |
| **YTHDF3** | YTH domain family protein 3 | 77 | 3 | 2 |
| SRP54 | Signal recognition particle 54-kDa protein | 66 | 6 | 4 |
| **GNL2** | Nucleolar GTP-binding protein 2 | 55 | 4 | 3 |
| SCAPER | S phase cyclin A-associated protein in the endoplasmic reticulum | 54 | 9 | 5 |
| PLA2G4A | Cytosolic phospholipase A2 | 53 | 4 | 4 |

**TABLE 1** (Continued)

| Host protein type and gene name[a] | Description | Prot score | No. of biotinylated peptides | No. of unique peptides |
|---|---|---|---|---|
| DCP1B | mRNA-decapping enzyme 1B | 51 | 4 | 3 |
| AGL | Glycogen debranching enzyme | 50 | 10 | 4 |
| **JAKMIP3** | Janus kinase and microtubule-interacting protein 3 | 47 | 12 | 5 |
| NACAD | NAC-alpha domain-containing protein 1 | 44 | 14 | 4 |
| DDX24 | ATP-dependent RNA helicase DDX24 | 40 | 15 | 11 |

[a]Boldface indicates genes encoding proteins that interacted with both 5′-UTR RNA and 3′-UTR RNA.

significantly higher than that predicted through random probabilities and/or known interactions within the reference genome (*Homo sapiens* for our analysis). 5′-UTR and 5′-UTR + 3′-UTR interaction RPPI networks showed significant enrichment of interactions (5′-UTR RPPI network characteristics: observed number of edges, 27; expected number of edges, 10; number of nodes, 46; average node degree, 1.17; average clustering coefficient, 0.311; PPI enrichment *P* value, 3.64e−06) (5′-UTR + 3′-UTR RPPI network characteristics: observed number of edges, 39; expected number of edges, 14; number of nodes, 56; average node degree, 1.39; average clustering coefficient, 0.29; PPI enrichment *P* value, 1.27e−08)]. Note that based on the above network parameters, the 5′-UTR + 3′-UTR RPPI network is highly connected in comparison to the 5′-UTR network alone (Fig. 2C; Fig. S3C).

Next, gene ontology (GO) and Reactome pathway analysis of the RPPI network was performed using the Gene Set Enrichment Analysis (GSEA) tool to determine the significantly enriched processes/pathways (Fig. S4 and S5) (28, 29). Of note, proteins involved in intracellular transport, translation initiation, amide biosynthetic pathway, and mRNA metabolism were enriched in the GO–biological processes category (Table 3). Similarly, proteins involved in cellular response to external stimuli, HSP90 chaperones for steroid hormone receptors, RNA metabolism, translation, influenza virus infection, infectious disease, and cell cycle, etc., were enriched in Reactome pathway analysis (Table 3). Since 5′ and 3′ UTRs are well conserved, drugs directly targeting the UTRs or acting upon the proteins that interact with the UTRs may function as potent antivirals by inhibiting the viral replication process.

Analysis of the 57 proteins identified in RaPID assay using the CyTarget Linker plug-in in Cytoscape software (30) as well as the Drug Gene Interaction Database (DGID) (31) identified 5 proteins against which known drugs exist. Of these 5 proteins, HABP4, TUB1A1, and PLA2G4A are reported to be targeted by hyaluronan, albendazole, and fluticasone propionate, respectively (32–35). NADH and tetrahydrofolic acid are cofactors for PDHA1 and MTHFD1, respectively (35) (Table 4).

**Lamp2 is a host restriction factor against SARS-CoV-2.** Lamp2a (CD107b) was identified as an interaction partner of the 5′-UTR RNA in the RaPID assay. LAMp2a is the receptor for chaperone-mediated autophagy (CMA), and recent studies have

**TABLE 2** Comparison of RaPID-identified host protein data set with other known data sets

| SARS-CoV-2 RNA-binding host proteins | Data sets compared from indicated studies | Genes encoding the proteins in common |
|---|---|---|
| Commonly identified in this study and reported by other independent studies | This study and that of Schmidt et al. (23) | G3BP2, ABCE1, RPL13, HSP90AA2, CTTN, RPL7A, SNRNP200, PDHA1, STIP1, HDLBP, TUBA1A |
| | This study and that of Flynn et al. (24) | MTHFD1, DIS3, PHB2, RPL7A, G3BP2, RPL13, SNRNP200, EIF5B, GEMIN5, MSH6, STIP1, STMN2, RANGAP1, HDLBP, POLDIP3, CDV3, FKBP4, UBA1, SRP54, AGL, RPL4, ABCE1, CTTN, TUBA1A1, YTHDF3 |
| Identified in this study, knockout of which significantly affects viral infection | This study and that of Daniloski et al. (25) | RPL13, TPR, SNRNP200, EIF5B, SLC25A31, AHNAK2 |
| | This study and that of Wei et al. (26) | TPR |
| | This study and that of Flynn et al. (24) | ABCE1, DIS3, G3BP2, GEMIN5, MTHFD1, RPL7A, SRP54, STIP1, TUBA1A |

**TABLE 3** Gene ontology and Reactome pathway analyses of SARS-CoV-2 5'-UTR and 3'-UTR RNA-protein interaction network[a]

| Gene set name | Observed gene count | P value | Gene names |
|---|---|---|---|
| **Gene ontology (biological process)** | | | |
| GO_INTRACELLULAR_TRANSPORT | 19 | 1.87E−12 | TPR, ABCE1, RPL7A, RPLP2, RPL13, RPL4, POLDIP3, SRP54, LAMP2, CTTN, PHB2, SNF8, HSPA1L, GTSE1, RANGAP1, NACAD, GEMIN5, TUBA1A, ARHGAP21 |
| GO_INTRACELLULAR_PROTEIN_TRANSPORT | 16 | 5.32E−12 | TPR, ABCE1, RPL7A, RPLP2, RPL13, RPL4, POLDIP3, SRP54, LAMP2, CTTN, PHB2, SNF8, HSPA1L, GTSE1, RANGAP1, NACAD |
| GO_TRANSLATIONAL_INITIATION | 9 | 8.13E−12 | TPR, ABCE1, RPL7A, RPLP2, RPL13, RPL4, YTHDF3, HABP4, EIF5B |
| GO_AMIDE_BIOSYNTHETIC_PROCESS | 14 | 2.51E−11 | TPR, ABCE1, RPL7A, RPLP2, RPL13, RPL4, POLDIP3, GEMIN5, YTHDF3, HABP4, EIF5B, MRPL40, PDHA1, MTHFD1 |
| GO_PROTEIN_CONTAINING_ COMPLEX_SUBUNIT_ORGANIZATION | 18 | 1.38E−10 | TPR, ABCE1, SRP54, LAMP2, CTTN, GEMIN5, MRPL40, HSPA1A, SNRNP200, G3BP2, STMN2, STMN1, FKBP4, H3-3A, CKAP2, NASP, H3C1, H3-4 |
| GO_MRNA_METABOLIC_PROCESS | 13 | 2.80E−10 | RPL7A, RPLP2, RPL13, RPL4, POLDIP3, GEMIN5, YTHDF3, HABP4, HSPA1A, SNRNP200, DCP1B, DIS3, TTF2 |
| GO_PEPTIDE_BIOSYNTHETIC_ PROCESS | 12 | 6.54E−10 | TPR, ABCE1, RPL7A, RPLP2, RPL13, RPL4, POLDIP3, GEMIN5, YTHDF3, HABP4, EIF5B, MRPL40 |
| GO_CELLULAR_AMIDE_METABOLIC_ PROCESS | 14 | 9.20E−10 | TPR, ABCE1, RPL7A, RPLP2, RPL13, RPL4, POLDIP3, GEMIN5, YTHDF3, HABP4, EIF5B, MRPL40, PDHA1, MTHFD1 |
| GO_CELLULAR_MACROMOLECULE_ LOCALIZATION | 17 | 1.32E−09 | TPR, ABCE1, RPL7A, RPLP2, RPL13, RPL4, POLDIP3, SRP54, LAMP2, CTTN, PHB2, SNF8, HSPA1L, GTSE1, RANGAP1, NACAD, SEPTIN9 |
| GO_PEPTIDE_METABOLIC_PROCESS | 12 | 4.67E−09 | TPR, ABCE1, RPL7A, RPLP2, RPL13, RPL4, POLDIP3, GEMIN5, YTHDF3, HABP4, EIF5B, MRPL40 |
| **Reactome pathway** | | | |
| REACTOME_CELLULAR_RESPONSES_TO_EXTERNAL_STIMULI | 14 | 1.40E−13 | HSPA1A, RPL4, RPL7A, RPL13, RPLP2, TPR, TUBA1A, FKBP4, HSPA1L, STIP1, H3C1, H3-3A, H3-4, RAD50 |
| REACTOME_METABOLISM_OF_RNA | 12 | 1.60E−10 | HSPA1A, RPL4, RPL7A, RPL13, RPLP2, TPR, DIS3, UTP3, GEMIN5, DCP1B, POLDIP3, SNRNP200 |
| REACTOME_HSP90_CHAPERONE_ CYCLE_FOR_STEROID_HORMONE_ RECEPTORS_SHR_ | 5 | 1.93E−08 | HSPA1A, TUBA1A, FKBP4, HSPA1L, STIP1 |
| REACTOME_INFLUENZA_INFECTION | 6 | 1.01E−07 | HSPA1A, RPL4, RPL7A, RPL13, RPLP2, TPR |
| REACTOME_TRANSLATION | 7 | 2.11E−07 | RPL4, RPL7A, RPL13, RPLP2, SRP54, EIF5B, MRPL40 |
| REACTOME_INFECTIOUS_DISEASE | 10 | 3.64E−07 | HSPA1A, RPL4, RPL7A, RPL13, RPLP2, TPR, TUBA1A, H3C1, RANGAP1, SNF8 |
| REACTOME_RRNA_PROCESSING | 6 | 4.86E−07 | RPL4, RPL7A, RPL13, RPLP2, DIS3, UTP3 |
| REACTOME_CELL_CYCLE | 9 | 5.64E−07 | RPL, TUBA1A, H3C1, H3-3A, H3-4, RAD50, RANGAP1, CEP43, GTSE1 |
| REACTOME_SRP_DEPENDENT_COTRANSLATIONAL_PROTEIN_TARGETING_TO_MEMBRANE | 5 | 6.08E−07 | RPL4, RPL7A, RPL13, RPLP2, SRP54 |
| REACTOME_EUKARYOTIC_ TRANSLATION_INITIATION | 5 | 8.19E−07 | RPL4, RPL7A, RPL13, RPLP2, EIF5B |

[a]Top 10 processes/pathways.

**TABLE 4** Known drugs targeting SARS-CoV-2 5′-UTR and 3′-UTR RNA-interacting host proteins

| Viral RNA binding proteins | Gene name | Drug name and reference |
|---|---|---|
| 5′-UTR RNA-interacting host proteins | HABP4 | Hyaluronan (33) |
| | PDHA1 | NADH (35) |
| | MTHFD1 | Tetrahydrofolic acid (35) |
| 3′-UTR RNA-interacting host proteins | TUBA1A | Albendazole (32) |
| | PLA2G4A | Fluticasone propionate (34) |

proposed a role of autophagy in SARS-CoV-2 pathogenesis (36). Hence, the significance of the interaction between 5′-UTR RNA and Lamp2a in the SARS-CoV-2 life cycle was investigated using a mammalian cell culture-based infection model of SARS-CoV-2.

A Vero E6 cell-based infection model of SARS-CoV-2 was established to investigate the role of Lamp2a in the SARS-CoV-2 life cycle. Productive infection of Vero E6 cells with SARS-CoV-2 was detected by immunofluorescence staining of the viral nucleocapsid (N) protein at 48 h postinfection using anti-N antibody (Fig. 3A). N staining was not observed in uninfected cells (Fig. 3A). Expression of N protein was also detected by Western blotting using anti-N antibody at 48 h postinfection (Fig. 3B, upper panel). An aliquot of the samples was immunoblotted with anti-GAPDH antibody to monitor equal loading of protein in uninfected and infected cells (Fig. 3B, lower panel). The level of viral RNA in the culture medium (from released virus) and inside the cells (intracellular) was then measured by quantitative real-time PCR (RT-qPCR) analysis of 48-h and 72-h SARS-CoV-2-infected samples by using a SYBR green-based method. RT-qPCR detection of RNA polymerase II (RP II) and RNase P (RP) served as a normalization control for intracellular and secreted RNA quantities, respectively. An increase in viral RNA level was observed in 72-h infected intracellular and secreted samples, compared to 48-h infected samples, indicating productive infection of Vero E6 cells with SARS-CoV-2 (Fig. 3C). Aliquots of the RNA from culture medium were also used in a TaqMan probe-based one-step RT-qPCR analysis using primers and probes recommended by the Centers for Disease Control and Prevention (CDC), USA, for detection of SARS-CoV-2 (Fig. 3D). SYBR green and TaqMan RT-qPCR methods showed comparable results, and hence, the SYBR green method was used in subsequent experiments. Collectively, these data demonstrate the establishment of the Vero E6 infection model of SARS-CoV-2 in our experimental setups.

Fluorescence *in situ* hybridization (FISH) was conducted to monitor the interaction between the 5′-UTRs of the viral genome and Lamp2 in SARS-CoV-2-infected Vero E6 cells. Lamp2 (green)- and 5′-UTR RNA-specific (red) probes colocalized (indicated by yellow), suggesting their interaction (Fig. 4A). Next, SMARTpool small interfering RNA (siRNA) targeting Lamp2 RNA (targets all Lamp2 isoforms, pan-Lamp2 siRNA, designated Lamp2 siRNA) was used to deplete Lamp2 protein in Vero E6 cells. Western blotting of Vero E6 whole-cell extract after 48 and 72 h of Lamp2 siRNA transfection demonstrated that siRNA was effective in reducing the total Lamp2 level by >90% in siRNA-transfected cells (Fig. 4B, upper panel). GAPDH was used as a loading control (Fig. 4B, lower panel). RT-qPCR of 72-h siRNA-transfected SARS-CoV-2-infected Vero E6 cells (analyzed 48 h postinfection) revealed an increase in viral RNA level both in the culture medium (Fig. 4C) and inside the cells (Fig. 4D). Western blot analysis of an aliquot of the whole-cell extracts using anti-N antibody showed increased nucleocapsid protein level in Lamp2 siRNA-treated SARS-CoV-2-infected cells, compared to that in nontargeting (NT) siRNA-treated SARS-CoV-2-infected cells (Fig. 4E, upper panel). Western blotting of the GAPDH protein level served as a loading control (Fig. 4E, lower panel).

Results obtained in Vero E6 cells were further verified in a human hepatoma (Huh7) cell-based infection model of SARS-CoV-2. Recent reports have shown the utility of

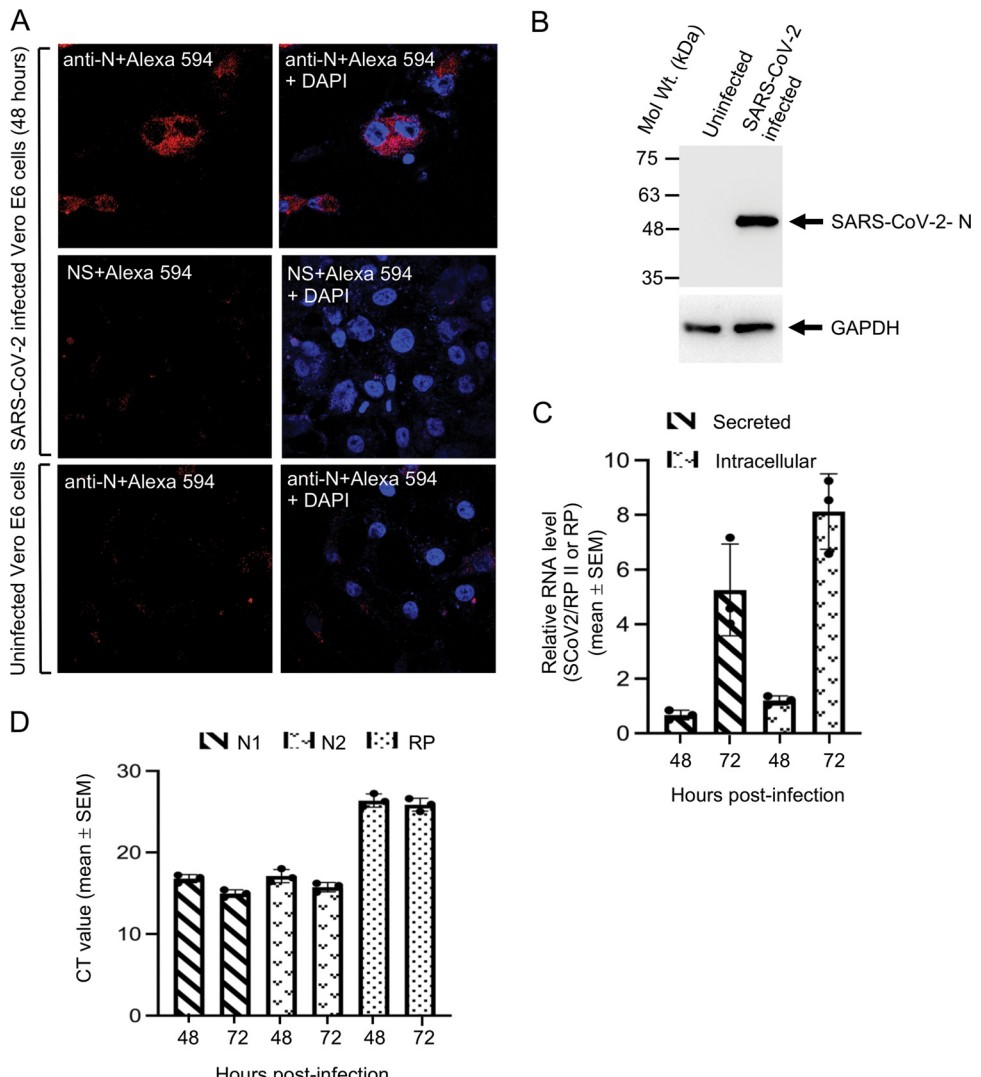

**FIG 3** Lamp2 associates with the 5' end of the SARS-CoV-2 genome and modulates the level of viral RNA in infected Vero E6 cells. (A) Immunofluorescence staining of nucleocapsid protein (red) and nucleus (blue) in SARS-CoV-2-infected Vero E6 cells, 48 h postinfection. Anti-N denotes incubation with nucleocapsid antibody, and NS denotes incubation with normal rabbit serum (preimmune serum control), followed by incubation with Alexa Fluor 594 conjugated secondary antibody. (B) Western blot detection of nucleocapsid protein (upper image) and GAPDH (lower image) in SARS-CoV-2-infected Vero E6 cells, 48 h postinfection. (C) Level of SARS-CoV-2 RNA in the culture medium (secreted, normalized to that of RNase P [RP]) and inside Vero E6 cells (intracellular, normalized to that of RNA Pol II [RP II]) that are infected with SARS-CoV-2 for the indicated periods. Real-time PCRs were performed using a SYBR green-based protocol. Data are the mean ± SEM of triplicate samples. (D). Level of SARS-CoV-2 RNA in Vero E6 cells in culture medium that are infected with SARS-CoV-2 for the indicated periods. Real-time PCRs were performed using a TaqMan probe-based protocol. N1, N2, and RP represents two regions within the SARS-CoV-2 nucleocapsid coding region and one region within RNase P (RP) that was selected for PCR amplification. Data are the mean ± SEM of triplicate samples. CT, threshold cycle.

Huh7 cells as a viable model for SARS-CoV-2 infection (23, 37). We observed the same effect in a Huh7 cell-based infection model of SARS-CoV-2 (Fig. S6A to S6F).

Lamp2 is present in three different forms in mammalian cells, including Lamp2a, Lamp2b, and Lamp2c, produced through alternative splicing (38). Lamp2a and Lamp2b are the major forms. Lamp2a is highly expressed in lungs, liver, and placenta. Lamp2b varies from Lamp2a in the last 11 amino acids of its C-terminal sequence, and it is highly expressed in skeletal muscle (39). Lamp2 antibody used in this study recognizes all three Lamp2 variants.

Next, we tested if a particular Lamp2 isoform is responsible for increased viral replication in Lamp2 siRNA-treated cells by designing siRNA against each isoform of

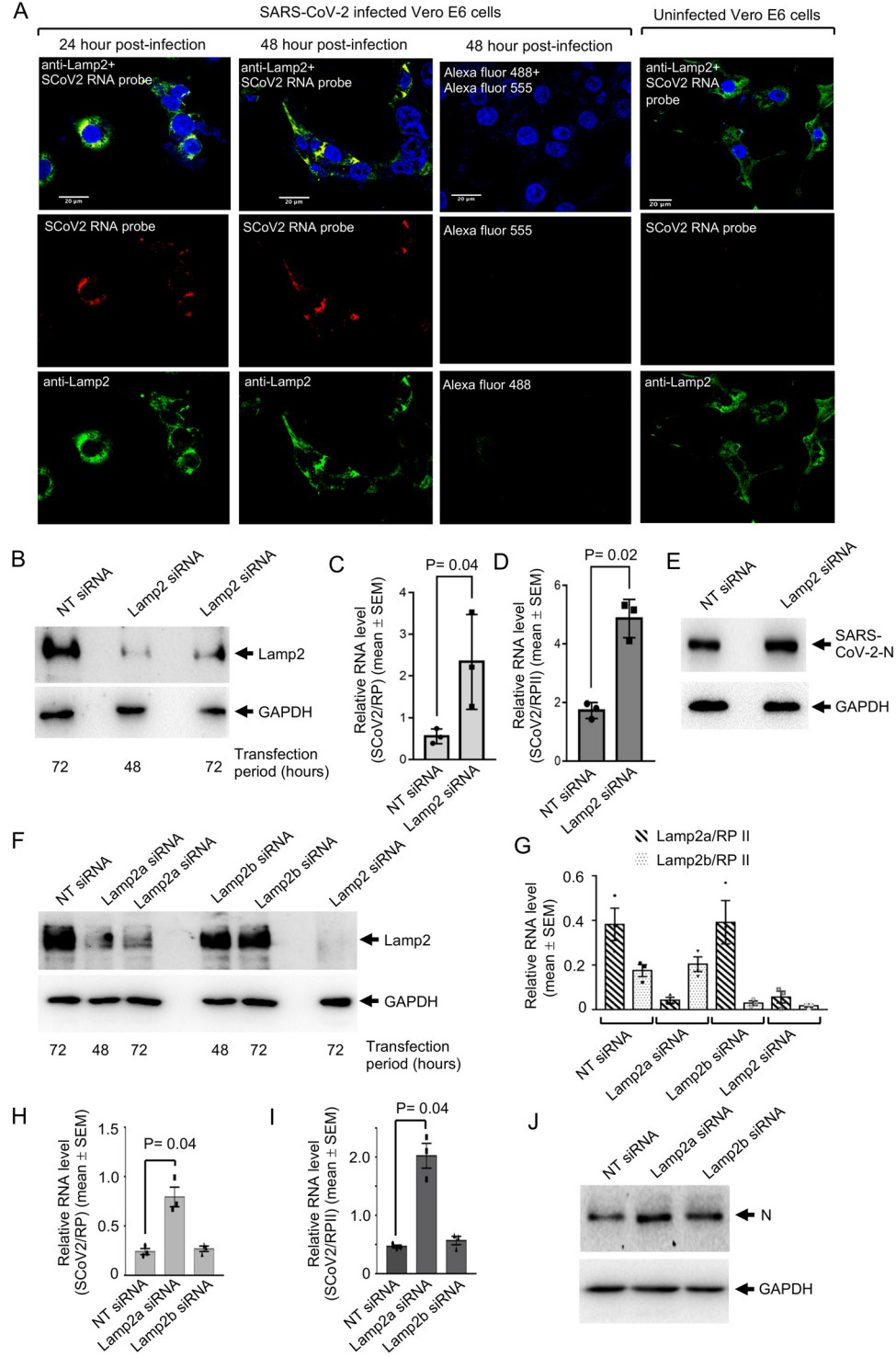

**FIG 4** Silencing of Lamp2a promotes SARS-CoV-2 infection in mammalian cell culture. (A) Fluorescence *in situ* hybridization of SARS-CoV-2 5'-UTR RNA and Lamp2, at indicated periods postinfection. Lamp2 protein, SARS-CoV-2 RNA, and nucleus are denoted by green, red, and blue, respectively. Scale bar, 20 μM. (B) Western blot detection of Lamp2 (Lamp2 antibody, upper panel) and GAPDH (lower panel) proteins in Vero E6 cells transfected for 72 h with nontargeting (NT) siRNA or Lamp2 siRNA. (C) Level of SARS-CoV-2 RNA (normalized to that of RP) in culture medium of Vero E6 cells transfected with NT siRNA or pan-Lamp2 siRNA (Lamp2 siRNA) and infected with SARS-CoV-2 for 48 h. Data are the mean ± SEM. *P* value was calculated using a two-tailed Student *t* test. (D) Intracellular level of SARS-CoV2 RNA (normalized to that of RP II) in Vero E6 cells treated with pan-Lamp2 siRNA (Lamp2 siRNA) and infected with SARS-CoV-2 for 48 h. Data are the mean ± SEM. *P* value was calculated using a two-tailed Student *t* test. (E) Western blot detection of nucleocapsid protein (N, upper image) and GAPDH (lower image) in nontargeting (NT) or pan-Lamp2 (Lamp2) siRNA-treated SARS-CoV-2-infected Vero E6

Lamp2. Analysis of Lamp2 sequence in multiple siRNA designing platforms indicated that specific siRNAs could be designed only against the Lamp2a and Lamp2b isoforms. siRNAs against human Lamp2a and Lamp2b were synthesized (details provided in Materials and Methods), and their silencing efficacy was tested in Huh7 cells. Western blot analysis of Lamp2a, Lamp2b, and pan-Lamp2 siRNA-treated Huh7 cell whole-cell extract revealed that both Lamp2a and pan-Lamp2 siRNAs significantly reduced the total Lamp2 protein level at both 48 and 72 h of treatment. Lamp2b siRNA treatment reduced the total Lamp2 protein level to a much lesser extent (Fig. 4F). In order to confirm the specificity of the Lamp2a and Lamp2b siRNAs, an RT-qPCR was performed in 72-h siRNA-treated samples using primers specific to both isoforms. As expected, isoform-specific siRNAs specifically reduced the level of corresponding RNA whereas pan-Lamp2 siRNA (designated Lamp2 siRNA) effectively reduced the level of both Lamp2a and Lamp2b isoforms (Fig. 4G). It was also clear that both Lamp2a and Lamp2b siRNAs were effective in reducing the Lamp2a and Lamp2b RNA levels by >80% (Fig. 4G) and that Lamp2a is the dominant isoform in Huh7 cells. Next, Huh7 cells were treated with Lamp2a and Lamp2b siRNAs, followed by infection with SARS-CoV-2 and measurement of viral replication by RT-qPCR. Only Lamp2a siRNA-treated samples showed an increased level of viral RNA both in the culture medium and at the intracellular level (Fig. 4H and I). Western blot analysis of a whole-cell extract of aliquots of the same sample showed an increase in the viral N protein level in Lamp2a siRNA-treated cells (Fig. 4J).

Next, the effect of overexpression of each Lamp2 isoform on SARS-CoV-2 replication was tested. Overexpression of Lamp2a and Lamp2b significantly reduced the viral RNA level, whereas Lamp2c overexpression did not alter viral the RNA level at 48 h postinfection, both in culture medium (Fig. 5A) and in intracellular samples in Vero E6 cells (Fig. 5B). Lamp2a, Lamp2b, and Lamp2c overexpression was confirmed in aliquots of the sample using anti-lamp2 antibody, which recognizes all Lamp2 variants (Fig. 5C to E), and using anti-HA antibody (for Lamp2a). Similar results were obtained in the Huh7 cell-based model of SARS-CoV-2 infection (Fig. S6G to S6K). Note that overexpressed Lamp2a and Lamp2b could be detected using both anti-HA and anti-Lamp2 antibodies in Huh7 cells but overexpressed Lamp2b could be detected using only anti-Lamp2 antibody in Vero E6 cells. Also note that no epitope tag is present in the Lamp2c expression clone. Hence, it was detected using anti-Lamp2 antibody.

In order to test if the observed reduction in viral RNA levels were specific to Lamp2, the effect of Lamp1 depletion on SARS-CoV-2 replication was evaluated. Western blot analysis using anti-Lamp1 antibody showed ~90% reduction in Lamp1 protein level in siRNA-transfected Huh7 cells at 48-h and 72-h time points (Fig. 5F). RT-qPCR analysis of Lamp1 siRNA-treated SARS-CoV-2-infected cells did not show any significant change in the level of viral RNA in culture medium (Fig. 5G) and intracellular samples (Fig. 5H).

**Increased Lamp2 protein level in SARS-CoV-2-infected cells does not promote chaperone-mediated autophagy or autophagolysosome formation.** Lamp2a is the receptor for chaperone-mediated autophagy (CMA), and silencing of Lamp2 expression promoted SARS-CoV-2 replication. GAPDH is known to be degraded by CMA, and a

**FIG 4 Legend (Continued)**
cells, 48 h postinfection. (F) Western blot detection of Lamp2 (upper image) and GAPDH (lower image) proteins in nontargeting (NT), Lamp2a, Lamp2b, or pan-Lamp2 (Lamp2) siRNA-treated Huh7 cell whole-cell extracts prepared at the indicated time points. (G) RT-qPCR measurement of Lamp2a and Lamp2b RNA levels (normalized to that of RP II) in Huh7 cells transfected for 72 h with nontargeting (NT), Lamp2a, Lamp2b, or pan-Lamp2 (Lamp2) siRNAs. Data are the mean ± SEM of triplicate samples. (H) RT-qPCR measurement of SARS-CoV-2 RNA levels (normalized to that of RP) in culture medium of Huh7 cells transfected for 72 h with nontargeting (NT), Lamp2a, or Lamp2b siRNAs and infected for 48 h with SARS-CoV-2. Data are the mean ± SEM of triplicate samples. P value was calculated using a two-tailed Student t test. (I) RT-qPCR measurement of intracellular level of SARS-CoV-2 RNA (normalized to that of RP II) in Huh7 cells transfected for 72 h with nontargeting (NT), Lamp2a, or Lamp2b siRNAs and infected for 48 h with SARS-CoV-2. Data are the mean ± SEM of triplicate samples. P value was calculated using a two-tailed Student t test. (J) Western blot detection of nucleocapsid (N) and GAPDH protein levels in Huh7 cells transfected for 72 h with nontargeting (NT), Lamp2a, or Lamp2b siRNAs and infected for 48 h with SARS-CoV-2.

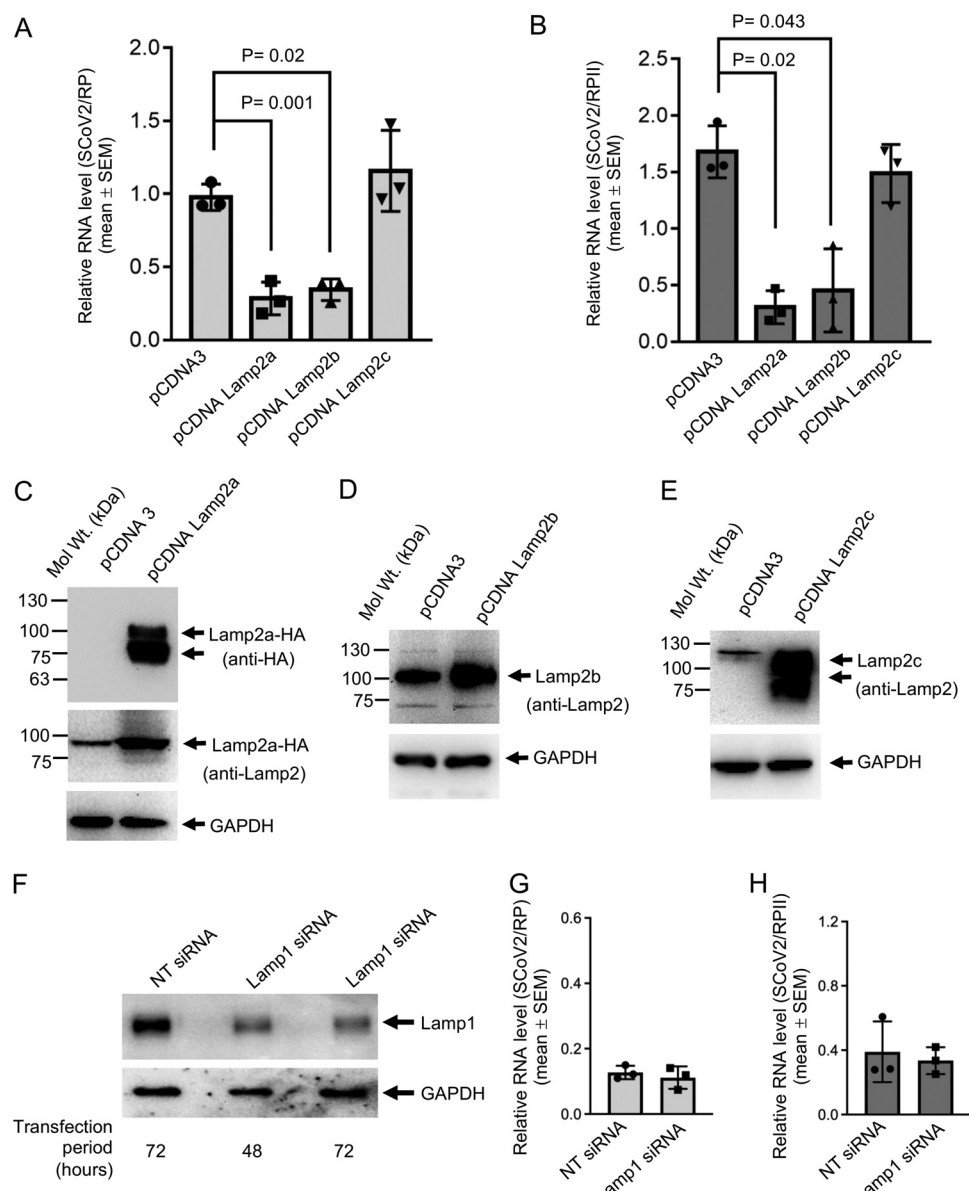

**FIG 5** Lamp2a modulates viral RNA level in SARS-CoV-2-infected cells. (A) SARS-CoV-2 RNA level (normalized to that of RP) in culture medium of Vero E6 cells transfected with the indicated plasmids and infected with SARS-CoV-2 for 48 h. Data are the mean ± SEM of triplicate samples. *P* values were calculated using a two-tailed Student *t* test. (B) Intracellular level of SARS-CoV-2 RNA (normalized to that of RP II) in Vero E6 cells transfected with the indicated plasmids and infected with SARS-CoV-2 for 48 h. Data are the mean ± SEM of triplicate samples. *P* values were calculated using a two-tailed Student *t* test. (C) Western blot detection of Lamp2a protein level (using anti-HA antibody [upper panel] and anti-Lamp2 antibody [middle panel]) and GAPDH protein level (lower panel) in pCDNA3-transfected or pcDNA Lamp2a-transfected Vero E6 Cells. (D) Western blot detection of Lamp2b protein level using anti-Lamp2 antibody (upper panel) and GAPDH protein level (lower panel) in pCDNA3-transfected or pcDNA Lamp2b-transfected Vero E6 cells. (E) Western blot detection of Lamp2c protein level using anti-Lamp2 antibody (upper panel) and GAPDH protein level (lower panel) in pCDNA3-transfected or pcDNA Lamp2c-transfected Vero E6 Cells. (F) Western blot detection of Lamp1 (upper panel) and GAPDH (lower panel) protein levels in Huh7 cells transfected for the indicated periods with nontargeting siRNA (NT) or Lamp1 siRNA. (G) Level of SARS-CoV-2 RNA (normalized to that of RP) in culture medium of Huh7 cells transfected for 72 h with Lamp1 siRNA and infected with SARS-CoV-2 for 48 h. Data are the mean ± SEM of triplicate samples. (H) Level of SARS-CoV-2 RNA (normalized to that of RP II) in culture medium of Huh7 cells transfected for 72 h with Lamp1 siRNA and infected with SARS-CoV-2 for 48 h. Data are the mean ± SEM of triplicate samples.

reduced GAPDH level is considered to be an indicator of CMA activation in a cell. An increase in the level of LAMP2 was observed; however, no significant change in GAPDH level was observed in SARS-CoV-2-infected Vero E6 or Huh7 cells, suggesting that CMA is not activated in these cells (Fig. 6A). We next measured the status of

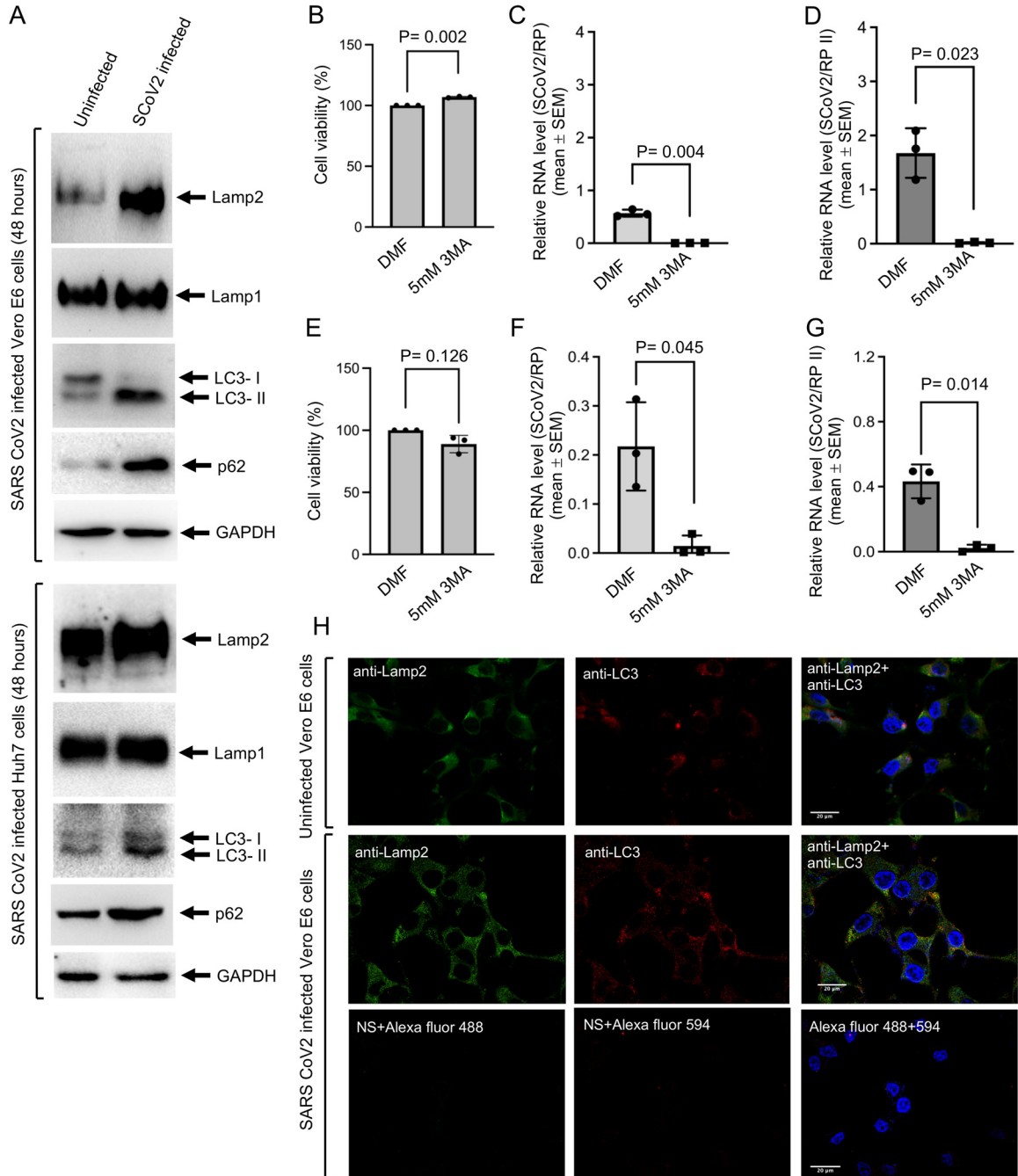

**FIG 6** Increased Lamp2 and LC3-II levels in SARS-CoV-2-infected Vero E6 and Huh7 cells do not promote autophagolysosome formation. (A) Western blot detection of indicated proteins in SARS-CoV-2-infected Vero E6 and Huh7 cells, 48 h postinfection. (B) Percent viability of Vero E6 cells treated for 48 h with 5 mM 3-MA or dimethylformamide (DMF) (vehicle). The value for the vehicle-treated sample was considered to be 100%, and other values were calculated with reference to that. Values are the mean ± SEM of triplicate samples. (C) Level of SARS-CoV-2 RNA (normalized to that of RP) in the culture medium of Vero E6 cells infected with SARS-CoV-2 and treated with 5 mM 3-MA or DMF (vehicle) for 48 h. Data are the mean ± SEM of triplicate samples. $P$ value was calculated using a two-tailed Student $t$ test. (D) Intracellular level of SRS-CoV-2 RNA (normalized to that of RP II) in Vero E6 cells treated with 5 mM 3-MA or DMF (vehicle) for 48 h. Data are the mean ± SEM of triplicate samples. $P$ value was calculated using a two-tailed Student $t$ test. (E) Percent viability of the Huh7 cells treated for 48 h with 5 mM 3-MA or DMF (vehicle). The value for the vehicle-treated sample was considered to be 100%, and other values were calculated with reference to that. Values are the mean ± SEM of triplicate samples. (F) Level of SARS-CoV-2 RNA (normalized to that of RP) in the culture medium of Huh7 cells infected with SARS-CoV-2 and treated with 5 mM 3-MA or DMF (vehicle) for 48 h. Data are the mean ± SEM of triplicate samples. $P$ value was calculated using a two-tailed Student $t$ test. (G) Intracellular level of SARS-CoV-2 RNA (normalized to that of RP II) in Huh7 cells treated with 5 mM 3-MA or DMF (vehicle) for 48 h. Data are the mean ± SEM of triplicate samples. $P$ value was calculated using a two-tailed Student $t$ test. (H) Immunofluorescence staining of Lamp2 (green), LC3 (red), and nucleus (blue) in uninfected or 48 h SARS-CoV-2-infected Vero E6 cells. Yellow indicates colocalization of the red and green signals. Scale bar, 20 $\mu$M.

macroautophagy in SARS-CoV-2-infected cells. An increase in the level of LC3 II and p62 protein was observed in SARS-CoV-2-infected Vero E6 and Huh7 cells (Fig. 6A). No change in the level of Lamp1 was observed (Fig. 6A). Treatment of SARS-CoV-2-infected Vero E6 and Huh7 cells with 3-methyladenine (3-MA), which is a specific inhibitor of autophagy, completely diminished viral RNA in both culture medium and intracellular samples (Fig. 6B to G). Cell viability measurement of aliquots of the same samples ensured that 3-MA treatment did not cause cytotoxicity (Fig. 6B and E). Immunofluorescence staining of Lamp2 (green) and LC3 (red) proteins was then performed in SARS-CoV-2-infected or uninfected Vero E6 cells. Formation of autophagolysosome is indicated by colocalization of Lamp2 and LC3, which forms yellow colored puncta in dual-stained cells (40). No increase in autophagolysosome formation was visible in SARS-CoV-2-infected Vero E6 cells (Fig. 6H).

## DISCUSSION

In this study, we identified 57 host proteins that interact with the 5′ and 3′ UTRs of the SARS-CoV-2 genomic RNA, combined those data sets with the PPI data set proposed to be involved in viral replication, and constructed an RPPI network, which is likely to be assembled during the replication of SARS-CoV-2. Interactions of 5′ and 3′ regulatory elements with virus-encoded and host proteins mediate their function. Since 5′ and 3′ regulatory elements are known to play indispensable roles in the replication process of RNA viruses, identification of the components of the above-mentioned RNA-protein complex paves the way for designing novel antiviral therapeutic strategies against SARS-CoV-2.

During the course of this study, Schmidt et al. and Flynn et al. identified a number of host proteins that bind to SARS-CoV-2 RNA in infected Huh7 cells using the RAP-MS and ChIRP-MS techniques, respectively (23, 24). In contrast, our study focused on the identification of host proteins that directly or indirectly associate with the 5′ and 3′ UTRs of SARS-CoV-2. The RaPID assay was used to capture the RNA-binding host proteins. The RaPID assay relies on *in vivo* biotinylation of proteins that bind to the RNA of interest, which is expressed as a chimeric RNA in fusion with BirA ligase-recognizing RNA motifs. Under conditions of biotin availability, the BirA ligase biotinylates all proteins in its close proximity (~10-nm range), including those associated with the RNA of interest. Thus, the RaPID assay detects both transient and stable interaction partners of an RNA under normal culture conditions. Since the RaPID protocol allows stringent washing of the pulldown samples, nonspecific binders are easily removed during the assay. Nonspecific proteins were further removed from the mass spectrometry data by using the following three controls: (i) subtraction of 5′- and 3′-UTR binding protein data sets from that of pRMB vector (which expresses the BirA-recognizing RNA motif), (ii) subtraction of 5′- and 3′-UTR binding protein data sets from that of pRMB HEV 3′ UTR (an unrelated RNA sequence derived from the 3′ UTR of HEV), and (iii) subtraction of 5′- and 3′-UTR binding protein data sets from that of a "no biotin treatment" control. However, it is noteworthy that the RaPID assay protocol used by us biotinylates the lysines exposed on the surface of proteins within close proximity and thus it has the inherent limitation that not all RNA binding proteins may be equally detected. Also, the specificity of the data may be further enhanced by using an additional control, which includes parallel processing of cells expressing pRMB 5′- and 3′-UTR RNAs in the absence of BirA ligase expression. Nevertheless, comparison of our RaPID data set with the data obtained by two other RNA-protein interaction detection techniques (RAP-MS and ChIRP-MS) identified 27 common proteins. Almost 50% overlap of our data set with those from other independent studies supports the specificity of our assay. Our study identified an additional 30 host proteins that associate with 5′ and 3′ UTRs of SARS-CoV-2 RNA. This may be attributed to the ability of the RaPID technique to detect both stable and transient as well as direct and indirect (as a protein complex) RNA-protein interactions. The specificity of the data is evident from the fact that only one protein (HIST1H3) is common between SARS-CoV-2 and HEV RPPI data sets. Note that like

SARS-CoV-2, HEV is also a positive-strand RNA virus which contains a capped 5′ end, followed by a 5′ UTR and poly(A) tail preceded by a 3′ UTR (41). Therefore, some aspects of the replication process might be similar in the two viruses.

In addition to infecting lungs, SARS-CoV-2 has been reported to cause enteric infections. A high level of ACE2 is expressed in the intestinal epithelial cells, and COVID-19 patients suffer from gastrointestinal disorders (4). Therefore, we hypothesized that host proteins that are crucial for the survival of the virus are expressed in both lung and intestinal epithelium cells. A comparison of expression of 5′-UTR and 3′-UTR RNA binding proteins in lung and intestinal epithelium cells showed that 55 out of 57 proteins are expressed in both tissues, in support of the above statement.

Among the 5′- and 3′-UTR-interacting host proteins, HABP4, TUBA1A, and PLA2G4A are known to be targeted by hyaluronan, albendazole, and fluticasone propionate, respectively. Additionally, NADH and tetrahydrofolic acid act as cofactors for PDHA1 and MTHFD1, respectively. Further studies are required to demonstrate if these drugs play a role during SARS-CoV-2 infection.

Recently, a protein interaction map of SARS-CoV-2 has been reported (42). Since the RNA-protein complex assembled at the 5′ and 3′ UTRs and the viral RNA-dependent RNA polymerase (RdRp)-associated protein complex are known to drive the viral replication process in positive-strand RNA viruses, we combined the 5′-UTR + 3′-UTR RPPI data set and a previously reported PPI data set (only the subset of PPI proposed by Gordon and coworkers to be involved in viral replication) to generate an integrated RPPI network of SARS-CoV-2 replication machinery (see Fig. S7 in the supplemental material). As expected, the integrated RPPI network showed significantly higher PPI enrichment than the replication PPI alone, suggesting that in infected cells, there is close proximity between the 5′ and 3′ ends of the viral RNA and assembly of an RNA-protein complex consisting of viral 5′ UTR, 3′ UTR, virus-encoded proteins, and host proteins (Fig. S7A and B). PPIs among the 5′- and 3′-UTR RNA-interacting proteins likely bridge the 5′ and 3′ ends of the genome.

Gene ontology and Reactome pathway analysis of the SARS-CoV-2 5′- and 3′-UTR RPPI data sets show an enrichment of proteins involved in different processes, such as translation initiation, RNA metabolism, and infectious disease, which is expected. Therefore, the data obtained by the RaPID assay is a useful resource for future studies related to SARS-CoV-2. A comparison of SARS-CoV-2 5′-UTR and 3′-UTR-interacting proteins obtained in the RaPID assay with those shown in other positive-strand RNA viruses reveal a similar profile of many functional protein categories. Proteins belonging to the ribosome complex such as RPL7a, RPLP2, RPL13, and RPL4 and translation initiation factors such as eIF5B were found to interact with the SARS-CoV-2 5′ and 3′ UTR. Being a capped RNA, SARS-CoV-2 genomic RNA is supposed to utilize the canonical cap-dependent translation process for producing viral proteins, and the above-identified host factors might be used by the virus for efficiently translating its own RNA. The importance of host ribosomal proteins and translation regulators in the SARS-CoV-2 life cycle is supported by the finding that knockout of RPL7a, RPL13, and eIF5B significantly affected virus growth (25, 26). Heat shock proteins of the HSP70 and HSP90 family are required for maintenance of cellular homeostasis by virtue of their ability to recognize and fold or remove unfolded proteins (43). They are also important for proper assembly and/or disassembly of the viral replication complex (44). They require the adapter protein STIP1 for optimal activity. In the case of SARS-CoV-2, HSP71, HSP71L, and STIP1 were found to interact with the 5′ UTR. Dead box RNA helicases such as DDX1 and DHX15 are known to associate with the TRS in MHV (7). DDX24 was identified as interacting with the 3′ UTR of SARS-CoV-2. DDX24 is known to associate with RNA and negatively regulate RIG-I-like receptor signaling, resulting in inhibition of the host antiviral response (45). Several proteins encoded by SARS-CoV-2, such as nucleocapsid, ORF6, and ORF8, are known to inhibit the host antiviral response (46). Interaction of DDX24 with the SARS-CoV-2 3′ UTR might be an additional strategy

to inhibit the host antiviral response. Further studies are required to confirm the above hypothesis.

ATP-binding cassette subfamily E member 1 (ABCE1) is another important host protein that interacts with the 5′ UTR. ABCE1 (RNase L inhibitor) inhibits the activity of RNase L, which is activated by the host antiviral response mechanism in response to RNA virus infection or alpha/beta interferon (IFN-$\alpha/\beta$) stimulation (47). Active RNase L cleaves the viral RNA, which is prevented in the presence of ABCE1. It will be interesting to explore the significance of the 5′-UTR interaction with ABCE1. Note that ABCE1 was also identified as a direct interaction partner of SARS-CoV-2 RNA by both Schmidt et al. and Flynn et al. (23, 24). FKBP4 was also found to interact with the 5′ UTR. It is a member of the immunophilin protein family and binds to immunosuppressants FK506 and rapamycin (48). Furthermore, Ras GTPase activating protein binding protein 2 (G3BP2), which is key to stress granule formation, was found to interact with the 5′ UTR of SARS-CoV-2. SARS-CoV-2 RNA binding by G3BP2 has also been shown by Flynn et al., and knockout of G3BP2 using CRISPR technology revealed a proviral role of this protein in SARS-CoV-2-infected cells (24). G3BP1/2 has also been shown to bind to the N protein of SARS-CoV-2 (36). Stress granules play key roles in controlling viral infections (49). The N protein of SARS-CoV-2 is predicted to associate with the 5′ UTR. The functional significance of the interaction of both N and 5′ UTR with G3BP2 during the course of SARS-CoV-2 remains to be explored. The 5′ UTR was also found to interact with the exosome endoribonuclease RRP4, which is involved in cellular RNA processing and degradation. The RNA exosome complex is a quality control mechanism of the host that regulates mRNA turnover and degrades aberrant RNAs. SARS-CoV-2 might modulate this pathway for its own benefit.

The 5′ UTR was also found to interact with Lamp2a, which is another key protein involved in quality control processes of the host. Lamp2a is the cellular receptor for chaperone-mediated autophagy (CMA) (50). Activation of CMA leads to localization of Lamp2a in the lysosome, as it recognizes and carries the cargo to the lysosome for degradation. Although there was an increase in the Lamp2 level in SARS-CoV-2-infected Vero E6 and Huh7 cells, we did not detect CMA activation in the infected cells, which suggests that increased Lamp2a may be sequestered away from binding to the CMA cargo due to its interaction with the viral 5′ UTR. Our data show an increase in autophagic flux (macroautophagy) in SARS-CoV-2-infected cells and a significant reduction in viral RNA level by inhibition of autophagy. Therefore, it appears that increased autophagic flux promotes SARS-CoV-2 replication, as seen in many other RNA viruses (51). However, there was no change in autophagolysosome formation in SARS-CoV-2-infected cells. A similar phenomenon has been observed in the case of HCV infection (52). A lack of Lamp2 protein increased the level of viral RNA and protein in SARS-CoV-2-infected cells. Therefore, Lamp2a interaction with the 5′ UTR is unlikely to play a role in promoting viral replication.

Lamp2b and Lamp2c isoforms are known to directly interact with RNA and DNA, leading to RNautophagy/DNautophagy (53). Lamp2c is predominantly involved in importing nucleic acids to the lysosome for degradation (54). In order to test the role of different Lamp2 isoforms during SARS-CoV-2 infection, Lamp2a and Lamp2b isoform-specific siRNAs were used.

No siRNA could be designed against the Lamp2c isoform. However, pan-Lamp2 siRNA used in our study targets all three isoforms, as it recognizes the 5′ end of the Lamp2 transcript (see details in Materials and Methods). Note that Lamp2 isoforms differ in their C-terminal regions. Silencing of Lamp2a significantly reduced the level of total Lamp2 in Huh7 cells and increased the level of viral RNA, whereas its overexpression decreased the level of viral RNA in SARS-CoV-2-infected cells, suggesting its antiviral property. Silencing of Lamp2b moderately decreased the level of total Lamp2, but the level of viral RNA remained unaltered in SARS-CoV-2-infected cells. However, overexpression of Lamp2b reduced the level of viral RNA in infected cells. These differences might be attributed to the fact that Lamp2a, being the predominant isoform, maintains antiviral activity in the

absence of Lamp2b; however, overexpression of Lamp2b might be affecting viral infection directly/indirectly by inducing RNautophagy/DNautophagy. Therefore, Lamp2b may not have a role in SARS-CoV-2 infection at its endogenous level. Although Lamp2c is expressed at a low level in Huh7 cells, its role in SARS-CoV-2 infection could not be confirmed. The Lamp2a–5′-UTR interaction may be involved in delivering viral RNA to the endosome, for recognition by TLR7, the sensor for single-stranded RNA (ssRNA). The Lamp2a–5′-UTR interaction may also be involved in sequestering Lamp2a away from inducing chaperone-mediated autophagy. At the same time, increased autophagic flux provides energy and prevents apoptosis of infected cells, which provides a favorable environment for the virus to complete its life cycle. Further studies are required to understand the actual process. Nevertheless, the current study identifies the repertoire of host proteins that associate with SARS-CoV-2 5′ and 3′ UTRs and characterizes the role of host Lamp2a protein during SARS-CoV-2 infection.

## MATERIALS AND METHODS

**Plasmids and reagents.** Sequences corresponding to nucleotides 1 to 300 at the 5′ end (5′ UTR) and 29676 to 29900 at the 3′ end (3′ UTR) of the SARS-CoV2 isolate Wuhan-Hu-1 genome (GenBank accession no. NC_045512) were commercially synthesized and cloned into pUC57 vector (GenScript, NJ, USA). RNA motif plasmid cloning backbone (pRMB) (Addgene plasmid no. 107253; http://n2t.net/addgene:107253), BASU RaPID plasmid (Addgene plasmid no. 107250; http://n2t.net/addgene:107250), and RNA motif plasmid EDEN15 (Addgene plasmid no. 107252; http://n2t.net/addgene:107252) were gifted by Paul Khavari. pCDNA Lamp2b was a gift from Joshua Leonard (Addgene plasmid no. 71292; http://n2t.net/addgene:71292), and pCDNA lamp2c was a gift from Janice Blum (Addgene plasmid no. 89342; http://n2t.net/addgene:89342). pcDNA Lamp2a (catalog no. HG29846-CY) was purchased from Sino Biologicals (Beijing, China). Anti-Lamp2 (catalog no. 49067), anti-Lamp1 (catalog no. 9091), anti-LC3B (catalog no. 83506), and anti-P62 (catalog no. 8025s) antibodies were from Cell Signaling Technology (MA, USA). Anti-GAPDH antibody (catalog no. SC-25778) was from Santa Cruz Biotechnology (TX, USA). Anti-Flag antibody (catalog no. A190-101) was from Bethyl Laboratories (TX, USA). Anti-CUGBP1 antibody (catalog no. STJ92521) was from St John's Laboratory (London, UK). Goat anti-rabbit IgG–horseradish peroxidase (HRP) (catalog no. 4030-05) and goat anti-mouse IgG–HRP (catalog no. 1030-05) were from Southern Biotech (AL, USA). Goat anti-rabbit IgG–Alexa Fluor 488 (catalog no. A-11008), goat anti-mouse IgG–Alexa Fluor 647 (A-21235), and antifade gold with DAPI (P36931) were from Thermo Fisher Scientific (MA, USA). 3-MA (catalog no. M9281) and anti-HA antibody (catalog no. A190-101) were from Sigma (MO, USA). Nontargeting siRNA (catalog no. D-001810-10-20), pan-Lamp1 siRNA (catalog no. L-013481-02-0005), and pan-Lamp2 siRNA (catalog no. L-011715-00-0005) were from Dharmacon (CO, USA). siRNAs against human Lamp2a and Lamp2b isoforms were designed using the GenScript siRNA target finder tool and synthesized at Eurogentec (Liege, Belgium). The following sequences were selected: Lamp2a, 5′GGCAGGAGUACUUAUUCUAGU3′; Lamp2b, 5′GUGUUCGCUGGAUGAUGAC3′. A literature search revealed that similar sequences have been used earlier to specifically silence Lamp2a and Lamp2b isoforms in human cell lines (55, 56). However, the same siRNA sequence did not match Lamp2a and Lamp2b isoforms of *Chlorocebus aethiops*, and they were not effective in silencing the corresponding isoforms expressed in Vero E6 cells. Pan-Lamp2 siRNA was effective in both Huh7 and Vero E6 cells, as it targeted the 5′ end of the RNA, which is conserved across isoforms in both species. The following are the sequences of siRNAs present in the SMARTpool pan-Lamp2 siRNA: 5′CUCAAUAGCACCAUUA3′, 5′GCAUGUAUUUGGUUAAUGG3′, 5′GCAUUGGAACUUAAUUUGA3′, and 5′AAAUGCCACUUGCCUUUAU3′.

**Mammalian cell culture, transfection, and cell viability assay.** Vero E6 and HEK 293T cells were obtained from the ATCC (VA, USA). Huh7 cells have been described previously (57). Cells were maintained in Dulbecco's modified Eagle medium (DMEM) containing 10% fetal bovine serum (FBS) and 50 IU/ml penicillin and streptomycin in 5% $CO_2$. For plasmid transfection, cells were seeded at 70 to 80% confluence in DMEM plus 10% FBS and incubated overnight. The next day, the cells were transfected with the desired plasmid DNA using Lipofectamine 2000 transfection reagent (Life Technologies, CA, USA) at a 1:1 ratio, in accordance with the manufacturer's instruction. At 6 to 8 h posttransfection, culture medium was replaced with fresh DMEM plus 10% FBS.

For experiments involving siRNA-mediated gene silencing, Huh7 or Vero E6 cells were seeded at 70 to 80% confluence on 12-well tissue culture (TC) dishes and incubated overnight at 37°C, 5% $CO_2$. The next day, 25 nmol siRNA was transfected into each well using 0.35 $\mu$l DharmaFECT transfection reagent, in accordance with the manufacturer's instructions (Dharmacon, CO, USA). At 18 h posttransfection, culture medium was replaced with DMEM plus 10% FBS and the cells were maintained at 37°C, 5% $CO_2$, until further manipulation.

Cell viability was measured using a commercially available kit (CellTiter 96 AQ$_{ueous}$ One Solution cell proliferation assay; Promega, Madison, USA) that employs a tetrazolium salt-based colorimetric assay. Details are as described previously (57).

**RaPID assay.** 5′ UTRs and 3′ UTRs were PCR amplified from the pUC57 vector and cloned into RNA motif plasmid cloning backbone vector (pRMB) at the BsmBI restriction site using the following primers: 5′ UTR FP, ATTAAAGGTTTATACCTTCCCAGG; 5′ UTR RP, GTTTTCTCGTTGAAACCAGGG; 3′ UTR

FP, AATCTTTAATCAGTGTGTAACA; 3′ UTR RP, TTTTTTTTGTCATTCTCCTAAG. Positive clones were checked by restriction mapping and confirmed by DNA sequencing.

Production of 5′-UTR and 3′-UTR RNA from the pRMB constructs was verified by transfection of pRMB 5′-UTR and pRMB 3′-UTR plasmids along with pRMB vector into HEK293T cells, followed by RT-PCR detection of the 5′-UTR and 3′-UTR RNA using the above-mentioned primer pairs. Expression of BirA ligase in HEK293T cells upon transfection of BASU plasmid was verified by Western blotting (Fig. 1D). Time durations for biotin treatment were optimized by cotransfection of RNA motif plasmid EDEN15 (pRMB EDEN15) and BASU plasmid at a 6:1 ratio using Lipofectamine 2000 (1:1 ratio) into HEK293T cells. At 42 h posttransfection, 200 $\mu$M biotin was added to the medium and cells were incubated for 6, 12, and 18 h. At each time point, whole-cell extract was prepared in 2× Laemmli buffer and equal amounts of protein were resolved by 10% SDS-PAGE, followed by Western blotting using anti-biotin antibody (Fig. 1E).

For verifying the interaction between EDEN15 and CUGBP1 by the RaPID assay, HEK293T cells were transfected with pRMB EDEN15 and BASU plasmids (6:1 ratio) using Lipofectamine 2000 (1:1 ratio; Life Technologies, CA, USA). At 40 h posttransfection, culture medium was replaced and 200 $\mu$M biotin was added. At 18 h later, cells were washed three times in phosphate-buffered saline (PBS) and lysed in pre-chilled radioimmunoprecipitation (RIPA) buffer (150 mM NaCl, 1.0% NP-40, 0.5% sodium deoxycholate, 0.1% SDS, 50 mM Tris, pH 8.0) supplemented with protease and phosphatase inhibitor cocktail. The lysate was centrifuged at 14,000 × $g$ for 45 min at 4°C. Free biotin was removed by a Macrosep advance spin filter (3,000 molecular weight cutoff [MWCO], 20 ml; catalog no. 89131-974; VWR, USA). Aliquots of the samples were stored at −80°C for use as input in Western blotting. The protein concentration in each sample was determined using a bicinchoninic acid (BCA) protein assay kit (Thermo Fisher Scientific, MA, USA), and equal amounts of proteins were incubated with MyOne streptavidin C1 magnetic beads (catalog no. 65002; Life Technologies, CA, USA) on a rotator at 4°C for 16 h. Samples were washed four times with washing buffer, and 2× Laemmli buffer was added to the beads and incubated at 95°C for 45 min. Both input and pulldown samples were resolved by SDS-PAGE, followed by Western blotting using anti-CUGBP1 and anti-GAPDH antibodies. Goat anti-rabbit IgG–HRP conjugated secondary antibody was used to detect the protein bands by chemiluminescence (Clarity ECL substrate; catalog no. 170-5061; Bio-Rad, CA, USA).

For mass spectrometry, clarified cell lysate was precipitated in acetone at −20°C for 10 min, followed by storage at −80°C for 20 min. The precipitates were solubilized in 8 M urea. The protein concentration was estimated by using the BCA protein assay kit (Thermo Fisher Scientific, MA, USA).

**(i) In-solution digestion and peptide separation.** An equal amount of protein (10 mg) from each sample was treated with 10 mM dithiothreitol (DTT; 56°C, 30 min) and alkylated with 20 mM iodoacetamide (IAA) (at room temperature, for 60 min, in the dark). Trypsin (catalog no. T1426; Thermo Fisher Scientific, MA, USA) was added to the samples at a 1:20 (wt/wt) ratio and incubated at pH 8, 37°C, for 24 h. Next, 1% formic acid was added to the samples, and peptides were desalted using a Sep-Pak C$_{18}$ cartridge (catalog no. WAT020515; Waters, MA, USA) and subsequently lyophilized in a SpeedVac.

High-capacity streptavidin agarose resins (catalog no. 20361; Thermo Fisher Scientific, MA, USA) were used to pull down the biotinylated peptides. The beads were washed in binding buffer (50 mM Na$_2$HPO$_4$, 150 mM NaCl, pH 7.2) before use.

Lyophilized peptides were solubilized in 1 ml PBS and incubated with 150 $\mu$l washed streptavidin agarose beads for 2 h at room temperature. The beads were washed once in 1 ml PBS, once in 1 ml washing buffer (5% acetonitrile in PBS), and finally once in ultrapure water. Excess liquid was completely removed from the beads, and biotinylated peptides were eluted by adding 0.3 ml of a solution containing 0.1% formic acid and 80% acetonitrile in water by boiling at 95°C for 5 min. A total of 10 elutions were collected and dried together in a SpeedVac. Enriched peptides were desalted with C$_{18}$ tips (Thermo Fisher Scientific, MA, USA), and reconstituted with solvent A (2% [vol/vol] acetonitrile, 0.1% [vol/vol] formic acid in water) for LC-MS/MS analysis.

**(ii) LC-MS/MS acquisition.** LC-MS/MS experiments were performed using a Sciex 5600$^+$ triple-time-of-flight (TOF) mass spectrometer coupled with a ChromXP reversed-phase 3-$\mu$m C$_{18}$-CL trap column (350 $\mu$m by 0.5 mm, 120 Å; Eksigent; AB Sciex, MA, USA) and a nanoViper C$_{18}$ separation column (75 $\mu$m by 250 mm, 3 $\mu$m, 100 Å; Acclaim PepMap; Thermo Fisher Scientific, MA, USA) in an Eksigent nanoLC (ultra 2D plus) system. The binary mobile solvent system used was as follows: solvent C, 2% (vol/vol) acetonitrile, 0.1% (vol/vol) formic acid in water; solvent B, 98% (vol/vol) acetonitrile, 0.1% (vol/vol) formic acid. The peptides were separated at a flow rate of 200 nl/min in a 60-min gradient with a total run time of 90 min. The MS data of each condition were acquired in IDA (information-dependent acquisition) with high sensitivity mode. Each cycle consisted of 250- and 100-ms acquisition times for MS1 ($m/z$ 350 to 1,250 Da) and MS/MS (100 to 1,800 $m/z$) scans, respectively, with a total cycle time of ∼2.3 s. Each condition was run in triplicate.

**(iii) Protein identification and quantification.** All raw files (.wiff) were searched using ProteinPilot software (version 4.5; Sciex) with the Mascot algorithm for protein identification and semiquantitation against the Swiss-Prot_57.15 database (20,266 sequences after application of *Homo sapiens* taxonomy filter). The search parameters for identification of biotinylated peptides were as follows: (i) trypsin as a proteolytic enzyme (with up to two missed cleavages); (ii) peptide mass error tolerance of 20 ppm; (iii) fragment mass error tolerance of 0.20 Da; and (iv) carbamido-methylation of cysteine (+57.02146 Da), oxidation of methionine (+15.99492 Da), deamination of NQ (+0.98416), and biotinylation of lysine (+226.07759 Da) as variable modifications. The quality of data between different samples and replicates was monitored by a Pearson correlation plot of peptide intensity against each run.

**(iv) Data analysis.** Proteins with at least one corrected biotinylated peptide and a posterior error probability (PEP) score of ≥15 were considered to be identified successfully and extracted from the Gaussian smoothing curve (note that PEP score refers to the probability that the observed peptide spectrum matches are incorrect). Next, all data were analyzed using a Web-based tool, Bioinformatics & Evolutionary Genomics (http://bioinformatics.psb.ugent.be/webtools/Venn/) to generate the Venn diagram to identify proteins that are unique interaction partners of 5'-UTR and 3'-UTR RNA. The background-subtracted data set was sorted on the basis of the parameters of a minimum of 2 unique peptides and a "prot score" of 40 or more to generate the final list of proteins, which were considered for further studies (Table 1). Note that "prot score" refers to the overall protein score calculated by Mascot, taking into account all observed mass spectra which matches amino acid sequences within a particular protein. The confidence level of the protein match was ensured by setting a high "prot score" threshold.

**Bioinformatics analysis.** RNA secondary structure was analyzed using the mfold program (http://www.unafold.org/mfold/applications/rna-folding-form.php), based on a minimum free energy calculation at 25°C (58). The virus-host RPPI data set was visualized using Cytoscape (version 3.1.0) (27). NetworkAnalyzer plug-in in Cytoscape was used to compute the topological parameters and centrality measures of the network. The CyTarget Linker plug-in in Cytoscape and the Drug Gene Interaction Database (DGID) were used to search for drug targets (30, 31). Gene ontology (GO) and Reactome pathway analysis was performed using the Gene Set Enrichment Analysis tool (https://www.gsea-msigdb.org/gsea/index.jsp) (28, 29).

**SARS-CoV-2 infection.** SARS-CoV-2 was obtained from BEI Resources (NR-52281; SARS-related coronavirus 2 isolate ISA-WA1/2020), amplified in Vero E6 cells in the biosafety level 3 (BSL3) facility of THSTI, India, titrated, and stored frozen in aliquots. For SARS-CoV-2 infection studies, 200 $\mu$l of stock virus was diluted in serum-free medium to 2,000 50% tissue culture infective doses ($TCID_{50}$)/ml and added to a Vero E6/Huh7 cell monolayer (seeded in a 24-well plate) for 1 h at 37°C, supplemented with 5% $CO_2$. At 1 h postincubation, the infection medium was removed, cells were washed twice with 500 $\mu$l of serum-free medium, and fresh DMEM plus 10% FBS was added. In the case of plasmid overexpression or siRNA transfection study, cells were transfected with the respective DNA or siRNA 24 h prior to infection with SARS-CoV-2. In the case of 3-MA treatment, 5 mM (final concentration) 3-MA was added to the culture medium during infection, again added to the complete medium after removal of the infection medium, and maintained for 48 h, followed by collection of culture medium and cells and subsequent experiments.

**Preparation of RNA probe for fluorescence *in situ* hybridization (FISH).** Sequence corresponding to the 5' 300 bases of SARS-CoV-2 was PCR amplified from pUC57 vector and cloned into the pJet1.2 vector (Thermo Fisher Scientific, MA, USA) under the control of the T7 promoter. pJet1.2_5'-300 plasmid was linearized by restriction digestion with XbaI. DIG-11-UTP (catalog no. 11209256910; Roche, Basel, Switzerland)-labeled RNA was *in vitro* synthesized using a MAXIscript *in vitro* transcription kit (Thermo Fisher Scientific, MA, USA), in accordance with the manufacturer's instructions. Template DNA was removed by treatment with DNase I, followed by precipitation of RNA using LiCl. An aliquot of the probe was resolved by agarose gel electrophoresis to monitor its size and integrity.

**FISH.** Fluorescence *in situ* hybridization (FISH) was done as described previously, with minor modifications in which a Tyramide SuperBoost kit (catalog no. B40933; Thermo Fisher Scientific, MA, USA) was used to detect the fluorescence signal (59). In summary, Vero E6 cells were seeded at 70 to 80% confluence on a coverslip overnight. The next day, cells were infected with the SARS-CoV-2 (see "SARS-CoV-2 infection") and incubated for 24 or 48 h. Cells were then washed three times in PBS, followed by fixation in 4% paraformaldehyde for 10 min at room temperature. Cells were rehydrated in 2× SSC buffer (300 mM NaCl, 30 mM sodium citrate) for 10 min at room temperature, followed by incubation with 50% formamide in 2× SSC buffer for 30 min at room temperature. Coverslips were then washed four times with warm hybridization buffer (10% formamide in 2× SSC, warmed to 70°C), followed by incubation with hybridization buffer containing 2 ng/$\mu$l of probe for 3 days at 42°C in a humified incubator. Cells were washed two times with 4× SSC buffer for 10 min each, at 42°C, and incubated with 20 $\mu$g/ml RNase A in 2× SSC buffer for 30 min at 37°C. Cells were then washed once with 2× SSC buffer and once with 0.1× SSC buffer for 10 min at 42°C. Endogenous peroxidase activity was quenched by adding 50 $\mu$l 3% $H_2O_2$ solution (provided in the Tyramide SuperBoost kit) and incubating at room temperature for 1 h. Cells were then blocked in 4% bovine serum albumin (BSA) (in PBS) for 1 h at room temperature, followed by incubation with anti-LAMP2 (1:50 dilution) and biotin-conjugated anti-DIG antibody (1:200 dilution) prepared in 4% BSA in PBST (PBS plus 0.2% Tween 20) overnight at 4°C. The next day, the cells were washed three times in PBS (10 min each) and incubated with HRP-conjugated streptavidin (provided in the Tyramide SuperBoost kit) for 60 min at room temperature Next, the cells were washed three times in PBS and incubated with goat-anti rabbit IgG–Alexa Fluor 488 (1:1,000 dilution, prepared in 4% BSA plus PBS plus 0.1% Tween 20) for 1 h at room temperature. Cells were washed three times in PBS. Next, freshly prepared tyramide working solution was added to the cells and incubated for 5 min at room temperature, followed by the addition of stop solution (provided with the Tyramide SuperBoost kit) and incubation for 3 min. Next, cells were washed three times in PBS and coverslips were mounted on a slide using antifade gold. Images were acquired using a 100× objective in a confocal microscope (Olympus FV3000) and analyzed by Image lab Fiji software.

**RNA isolation and RT-qPCR assay.** Intracellular RNA was isolated using TRI reagent (MRC, MA, USA), followed by reverse transcription (RT) using a FIREScript cDNA synthesis kit (Solis Biodyne, Estonia). RNA from culture medium was isolated using a Qiagen viral RNA minikit (Qiagen, Germany), followed by RT using a FIREScript cDNA synthesis kit (Solis Biodyne, Estonia). Random hexamers were used in cDNA synthesis. SYBR green-based quantitative real-time PCR (RT-qPCR) was done as described earlier (57). The following primers were used: SCoV2 QPCR FP, 5'-TGGACCCCAAAATCAGCGAA; SCoV2 QPCR RP, 5'-

TCGTCTGGTAGCTCTTCGGT; RP II FP, 5′-GCACCACGTCCAATGACAT; RP II RP, 5′-GTCGGCTGCTTCCATAA; RP FP, 5′-AGATTTGGACCTGCGAGCG; RP RP, 5′-GAGCGGCTGTCTCCACAAGT; LAMP2A FP, 5′-TATGTGCAA CAAAGAGCAGAC3; LAMP2A RP, 5′-AAGCCAGCAACACTAGAATAAG3; LAMP2B FP: 5′-TATGTGCAACAAA GAGCAGAC3, LAMP2B RP, 5′-TGCCAATTACGTAAGCAATCA. TaqMan-based RT-qPCR was done as described earlier, by following the protocol suggested by the CDC, USA (60). The following primers and probes were used: N1 FP, 5′-GACCCCAAAATCAGCGAAAT; N1 RP, 5′-TCTGGTTACTGCCAGTTGAATCTG; N1 probe, 5′-FAM-ACCCCGCATTACGTTTGGTGGACC-BHQ1; N2 FP, 5′-TTACAAACATTGGCCGCAAA; N2 RP, 5′-GCGCGACATTCC GAAGAA; N2 probe, 5′-FAM-ACAATTTGCCCCCAGCGCTTCAG-BHQ1; RP FP, 5′-AGATTTGGACCTGCGAGCG; RP RP, 5′-GAGCGGCTGTCTCCACAAGT; RP probe, 5′-FAM-TTCTGACCTGAAGGCTCTGCGCG-BHQ1. Absolute quantification was used in SYBR green RT-qPCRs. A standard plot was generated by serial dilution of a known quantity of template. SARS-CoV-2 PCR values were normalized to that of RNA polymerase II (RP II, in-tracellular RNA) or RNase P (RP, culture medium RNA).

**Statistics.** Data are represented as the mean ± standard error of the mean (SEM) of three experi-ments. $P$ values were calculated using a two-tailed Student $t$ test (paired two samples for means).

**Immunofluorescence assay (IFA).** Confocal imaging was performed as described previously (57). In brief, Huh7 and Vero E6 cells were fixed with 4% paraformaldehyde for 10 min at room temperature, fol-lowed by incubation in blocking buffer (4% BSA in PBS) for 1 h at room temperature and incubation with Lamp2 and LC3b antibodies (1:50 and 1:50 dilutions, respectively) or anti-N antibody (1:50 dilution) in antibody dilution buffer (3% BSA in PBS plus 0.1% Tween 20) for 16 h, at 4°C. Coverslips were washed three times in PBS, followed by incubation with a 1:500 dilution of goat anti-rabbit–Alexa Fluor 488 and a 1:500 dilution of goat anti-mouse–Alexa Fluor 647 antibodies (Thermo Fisher Scientific, MA, USA) or goat anti-rabbit–Alexa Fluor 594 antibody (for anti-N detection; Thermo Fisher Scientific, MA, USA) in antibody dilution buffer at room temperature for 1 h. Coverslips were washed three times in PBS and mounted on glass slides using antifade gold reagent (Thermo Fisher Scientific, MA, USA). Images were acquired using a 60× objective in a confocal microscope (Olympus FV3000) and analyzed by Image lab Fiji software.

**Western blot assay.** Samples were resolved by SDS-PAGE, transferred to a 0.4-$\mu$m polyvinylidene diflu-oride (PVDF) membrane. Membranes were blocked for 1 h at room temperature using 5% skimmed milk (in PBS). Next, membranes were incubated overnight with primary antibody in PBST (PBS plus 0.05% Tween 20) plus 5% skimmed milk at 4°C. Blots were washed 3 times in PBST, followed by incubation with HRP-tagged secondary antibody at room temperature for 1 h. Blots were washed 3 times in PBST, and protein bands were visualized by enhanced chemiluminescence using a commercially available kit (Bio Rad, CA, USA).

**Data availability.** The mass spectrometry proteomics data are available in the MassIVE repository, which is a part of ProteomeXchange, under identifier number PXD026754. The data can be accessed directly at http://proteomecentral.proteomexchange.org/cgi/GetDataset?ID=PXD026754.

## SUPPLEMENTAL MATERIAL

Supplemental material is available online only.

**FIG S1**, PDF file, 0.1 MB.
**FIG S2**, PDF file, 0.1 MB.
**FIG S3**, PDF file, 1.1 MB.
**FIG S4**, PDF file, 1.6 MB.
**FIG S5**, PDF file, 1.5 MB.
**FIG S6**, PDF file, 0.2 MB.
**FIG S7**, PDF file, 0.2 MB.
**TABLE S1**, PDF file, 0.1 MB.
**TABLE S2**, PDF file, 0.04 MB.
**TABLE S3**, PDF file, 0.02 MB.

## ACKNOWLEDGMENTS

The RNA motif plasmid cloning backbone (pRMB), BASU RaPID plasmid, and RNA motif plasmid EDEN15 were kindly gifted by Paul Khavari. pCDNA Lamp2b was a gift from Joshua Leonard, and pCDNA Lamp2c was a gift from Janice Blum. SARS-related coronavirus 2, isolate USA-WA1/2020 (NR-52281), was deposited by the Centers for Disease Control and Prevention and obtained through BEI Resources, NIAID, NIH, USA. We thank Tripti Shrivastava for generously providing the rabbit polyclonal anti-N antibody (Genetex, Inc., USA).

This study was funded by the Science and Engineering Research Board (SERB), Government of India, by an IRHPA grant (IPA/2020/000233) to M.S. and a THSTI core grant to M.S. R.V. and S.S. are supported by a senior research fellowship from the Council of Scientific and Industrial Research, Government of India, and S.K. is supported

by a senior research fellowship from the Department of Biotechnology, Government of India.

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
