## [Reviewer comments · mSystems]

RNA-protein interaction analysis of SARS-CoV-2 5'- and 3'-untranslated regions reveal a role of lysosome-associated membrane protein-2a during viral infection

Rohit Verma, Sandhini Saha, Shiv Kumar, Shailendra Mani, Tushar Maiti, and Milan Surjit

Corresponding Author(s): Milan Surjit, Translational Health Science and Technology Institute

Review Timeline:

Submission Date:	May 27, 2021
Editorial Decision:	June 14, 2021
Revision Received:	June 17, 2021
Accepted:	June 22, 2021

Editor: Marta Gaglia

Reviewer(s): The reviewers have opted to remain anonymous.

Transaction Report:

DOI: <https://doi.org/10.1128/mSystems.00643-21>

June 14, 2021

Dr. Milan Surjit
Translational Health Science and Technology Institute
Faridabad
India

Re: mSystems00643-21 (RNA-protein interaction analysis of SARS-CoV-2 5'- and 3'-untranslated regions reveal a role of lysosome-associated membrane protein-2a during viral infection)

Dear Dr. Milan Surjit:

Thank you for submitting your manuscript to mSystems. We have completed our review and I am pleased to inform you that, in principle, we expect to accept it for publication in mSystems.

However, acceptance will not be final until you have made the mass spectrometry data publicly available through an appropriate database. Please refer to the ASM Journals policy at <https://journals.asm.org/availability-materials-and-data> for examples of suitable repositories. Also, you will need to add a paragraph to the methods section called "Data availability" detailing where the data can be downloaded from, including accession numbers per ASM Journals open data policy (<https://journals.asm.org/open-data-policy>).

Preparing Revision Guidelines

For complete guidelines on revision requirements, please see the Instructions to Authors at <https://msystems.asm.org/sites/default/files/additional-assets/mSys-ITA.pdf>. **Submissions of a paper that does not conform to mSystems guidelines will delay acceptance of your manuscript.**

Corresponding authors may join or renew ASM membership to obtain discounts on publication fees.

Need to upgrade your membership level? Please contact Customer Service at Service@asmusa.org.

Sincerely,

Marta Gaglia

Editor, mSystems

Journals Department
Reviewer comments:

Reviewer #1 (Comments for the Author):

Thank you for addressing the reviewers concerns.

June 22, 2021

Dr. Milan Surjit
Translational Health Science and Technology Institute
Faridabad
India

Re: mSystems00643-21R1 (RNA-protein interaction analysis of SARS-CoV-2 5'- and 3'-untranslated regions reveal a role of lysosome-associated membrane protein-2a during viral infection)

Dear Dr. Milan Surjit:

Your manuscript has been accepted, and I am forwarding it to the ASM Journals Department for publication. For your reference, ASM Journals' address is given below. Before it can be scheduled for publication, your manuscript will be checked by the mSystems senior production editor, Ellie Ghatineh, to make sure that all elements meet the technical requirements for publication. She will contact you if anything needs to be revised before copyediting and production can begin. Otherwise, you will be notified when your proofs are ready to be viewed.

We recognize that the video files can become quite large, and so to avoid quality loss ASM

suggests sending the video file via <https://www.wetransfer.com/>. When you have a final version of the video and the still ready to share, please send it to Ellie Ghatineh at eghatineh@asmusa.org.

Sincerely,

Marta Gaglia
Editor, mSystems

Journals Department
Fig. S6: Accept
Fig. S7: Accept
Table S3: Accept
Table S2: Accept
Fig. S3: Accept
Fig. S2: Accept
Fig. S4: Accept
Table S1: Accept
Fig. S1: Accept
Fig. S5: Accept